# Machine learning enables design automation of microfluidic flow-focusing droplet generation

Ali Lashkaripour 1,2, Christopher Rodriguez3,8, Noushin Mehdipour 2,4,8, Rizki Mardian2,7,8, David McIntyre 1,2,8, Luis Ortiz2,5, Joshua Campbell 6 & Douglas Densmore 2,7✉

Droplet-based microfluidic devices hold immense potential in becoming inexpensive alternatives to existing screening platforms across life science applications, such as enzyme discovery and early cancer detection. However, the lack of a predictive understanding of droplet generation makes engineering a droplet-based platform an iterative and resource-intensive process. We present a web-based tool, DAFD, that predicts the performance and enables design automation of flow-focusing droplet generators. We capitalize on machine learning algorithms to predict the droplet diameter and rate with a mean absolute error of less than 10 $\mu$m and 20 Hz. This tool delivers a user-specified performance within 4.2% and 11.5% of the desired diameter and rate. We demonstrate that DAFD can be extended by the community to support additional fluid combinations, without requiring extensive machine learning knowledge or large-scale data-sets. This tool will reduce the need for microfluidic expertise and design iterations and facilitate adoption of microfluidics in life sciences.

[1] Department of Biomedical Engineering, Boston University, Boston, MA, USA. [2] Biological Design Center, 610 Commonwealth Avenue, Boston, MA, USA. [3] Computational and Systems Biology, Massachusetts Institute of Technology, Cambridge, MA, USA. [4] Division of Systems Engineering, Boston University, Boston, MA, USA. [5] Department of Molecular Biology, Cell Biology & Biochemistry, Boston University, Boston, MA, USA. [6] Department of Medicine, Boston University, Boston, MA, USA. [7] Department of Electrical and Computer Engineering, Boston University, Boston, MA, USA. [8] These authors contributed equally: Christopher Rodriguez, Noushin Mehdipour, Rizki Mardian, David McIntyre. ✉email: dougd@bu.edu

Miniaturization of liquid handling is in ever increasing demand as the need for higher sensitivity and throughput rises in numerous life science applications, such as drug discovery, biochemical assays, clinical diagnostics, and genomics[1–4]. Robotic liquid handling[5], digital microfluidics[6], and droplet microfluidics[7] are commonly used to minimize sample volumes. Large machine footprints and high overhead costs limit the accessibility of high-performance liquid handling robots and cost-effective robots often operate at volumes larger than 1 µl[8]. Digital microfluidic devices, although ideal for running complex protocols[9], fall short of delivering a high throughput[10]. Conversely, droplet microfluidics enables unprecedented combination of throughput, volume reduction, reaction control, and sensitivity[11–13], yet, adoption of droplet-based platforms in the life sciences has been an exception rather than the norm[14,15]. This can be attributed to the phenomenological complexity of droplet formation[16,17], lack of predictive understanding[18–20], high fabrication cost inherent to photolithography[21], and unreliability of numerical simulations to capture the intricate dynamics of multiphase flows[20]. Consequently, expertise and an iterative design process is required to achieve the desired performance[22,23]. Several geometries including T-junction[7], step-emulsification[24], co-flow[25], and flow-focusing[26] can be used to generate droplets. Flow-focusing geometries offer a wider range of deliverable performance (i.e., droplet diameter and generation rate) in comparison to the other geometries[27–29]. Nonetheless, due to the large number of effective parameters and the complex fluid dynamics involved, analytical solutions or generalizable scaling laws are yet to be determined for flow-focusing droplet generation[18,20].

With the recent introduction of high-resolution low-cost rapid prototyping techniques, the barrier to entry to microfluidics is significantly lowered[30,31]. Large design spaces previously studied through numerical simulations[32,33] can now be explored experimentally to characterize the performance of droplet generators at a realistic cost and time-frame[23]. Therefore, sufficiently large data-sets can be generated to train machine learning algorithms and achieve accurate performance prediction, a need that has not been met since the introduction of droplet microfluidics almost two decades ago[18].

Machine learning enables detection of complex patterns using computer science and statistics[34]. With the increasing availability of large-scale data-sets, machine learning has helped advance numerous fields including cancer detection, cell behavior prediction, genomic analysis, and drug discovery[35–38]. However, implementing machine learning in the field of droplet microfluidics has been limited to real-time or post-experiment data analysis[39] due to the lack of standardized and sufficiently large data-sets[40]. The ability to predict the performance of droplet generators based on the design parameters eliminates costly design iterations and enables application-specific design optimization[18]. More importantly, accurate performance prediction allows design automation tools for droplet generators to be developed, significantly reducing the resources and expertise required to develop functional droplet-based platforms.

In this study, we leverage a low-cost rapid prototyping method[30], to fabricate 43 flow-focusing droplet generators, and evaluate their performance over a wide range of flow conditions, generating a standardized large-scale data-set of 998 data-points. Capitalizing on this data-set and machine learning algorithms, we develop a web-based tool, DAFD (Design Automation of Fluid Dynamics), that can predict the performance of droplet generators. We demonstrate DAFD can be extended to support additional fluid combinations through DAFD Neural Optimizer and transfer learning, providing a framework for machine learning based performance prediction of droplet generation with a diverse set of fluids. Furthermore, we develop and verify a

design automation tool that can design a droplet generator based on user-specified performance. Finally, we demonstrate that performance prediction enables further features, such as quantifying the effect of fabrication and testing tolerances on the observed performance, while providing a guideline to adjust flow rates to account for these possible tolerances. An overview of this study is shown in Fig. 1.

## Results

**Efficient large-scale data-set generation.** A microfluidic flow-focusing droplet generator can be defined with six geometric parameters: orifice width, orifice length, water inlet width, oil inlet width, outlet channel width, and channel depth (Fig. 1a). To capture the effect of geometry on droplet generation, these parameters were varied according to the range observed in the literature while considering fabrication limits (see Supplementary Note 2). Using a low-cost rapid prototyping and assembly technique we previously introduced[23,30], 43 flow-focusing devices were fabricated at fraction of time and cost required with standard photolithography (see Supplementary Note 1). 25 out of the 43 devices were designed to cover an orthogonal design space (using Taguchi design of experiments method, see Supplementary Table 2) as we previously described[23]. The remaining 18 devices were granularly added to the data-set during the verification process of the design automation tool until accurate design automation was achieved.

In addition to geometry, fluid properties and flow rates play major roles in dictating the behavior of microfluidic droplet generators. As shown in Fig. 2b, the fabricated devices were tested over a wide range of capillary number and flow rate ratio combinations (minimum 1 and maximum 34 different flow conditions per device), with a total of 65 unique flow conditions (not every device was tested at the same flow conditions to generate a diverse data-set). Nine hundred and ninety-eight experimental data-points were generated and droplet diameter, generation rate, and generation regime were recorded. The observed droplet diameter varied from 27.5 to 460 µm, generation rate range was observed to be 0.47–818 Hz, and the generation regime occurred in the dripping regime for 561 data-points and in the jetting regime for the remaining 437 data-points (see Fig. 2c).

**Performance prediction.** Flow-focusing droplet generation commonly occurs at dripping or jetting regimes, with significant differences in sensitivity to the design parameters and observed performance[23]. Consequently, generation regime prediction is integral to accurate performance prediction of droplet generators. To achieve this, neural network models were built, trained, and optimized to first, classify the generation regime, and then, predict the droplet diameter and generation rate. The regime classification model was built using all data-points and was able to predict droplet generation regime with an accuracy of 95.1 ± 1.5% against the test-set (20% of the data that the model was not trained on). The standard deviation is calculated based on ten different training–testing sessions, where the train-set and test-set were randomly chosen.

Droplet diameter, generation rate, and flow rate of the dispersed phase are interdependent and given two of the parameters the third parameter can be calculated using the conservation of mass principle as given Eq. (1):

$$\frac{1}{6}\pi D^3 \cdot F = Q_{\text{d}}, \tag{1}$$

where $D$ is droplet diameter, $F$ is generation rate, and $Q_{\text{d}}$ is the flow rate of the dispersed phase. Therefore, predicting either droplet diameter or generation rate is sufficient to calculate the

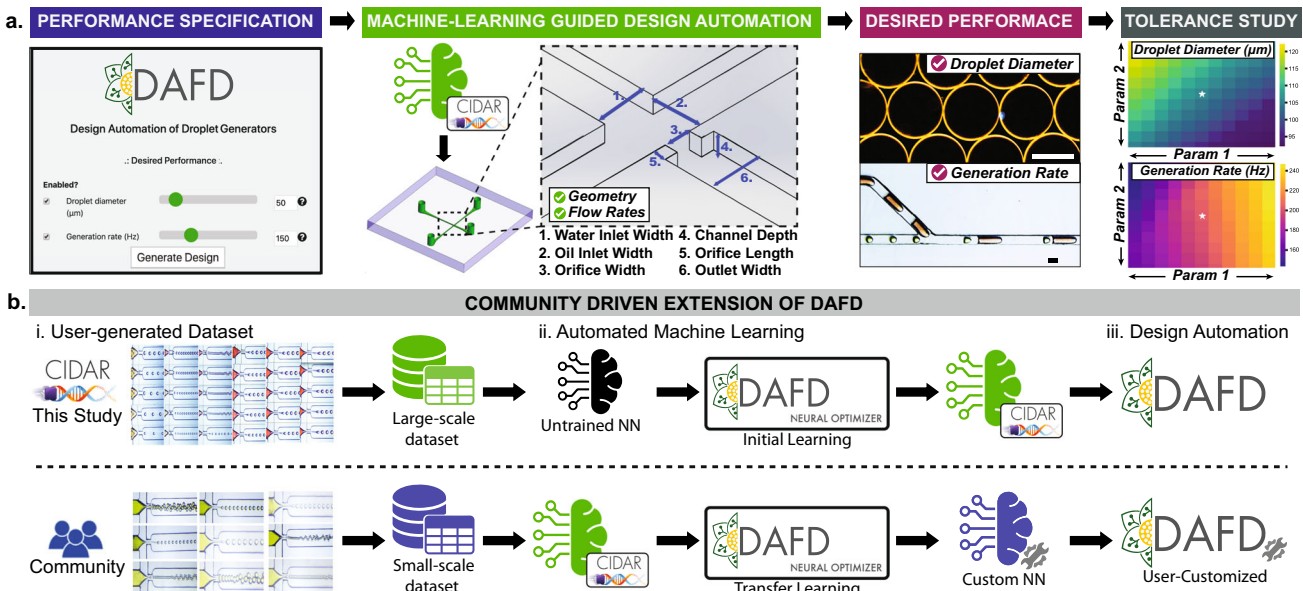

**Fig. 1 The workflow of the developed design automation tool for flow-focusing droplet generators, called DAFD.** This tool is made possible by accurate machine learning based predictive models developed in this study. **a** The machine learning algorithms convert the user-specified performance into the required geometry and flow rates to achieve the desired droplet diameter and generation rate. The developed tool can also predict performance deviations caused by possible tolerances in fabrication or testing. **b** Accurate predictive models (neural networks with CIDAR logo) trained on a large-scale data-set (initial learning) are made possible by machine learning and a low-cost rapid prototyping method that enabled efficient generation of a large-scale data-set in this study. Additional custom-built design automation tools that support further fluid combinations can be developed by the microfluidic community using Neural Optimizer to automatically train neural networks based on a user data-set without requiring extensive machine learning knowledge, or by using transfer learning and the pre-trained models (accurate predictive models developed in this study) without requiring a large-scale data-set. This tool is open-source and can be accessed online at: dafdcad.org. Scale bars represent 100 μm.

other, for a given flow rate. Nonetheless, here we developed separate models for predicting droplet diameter and generation rate to add redundancy in the design automation stage. This enabled defining a new parameter called "inferred droplet diameter" (the diameter calculated using the predicted generation rate and the conservation of mass principle), allowing for accuracy-checking of one predictive model using the other predictive model in order to avoid design-spaces where one or both models are inaccurate, as further explained in the design automation section.

Four neural network models for predicting droplet diameter and generation rate in the two generation regimes were built and trained on a bounded performance range (25–250 μm and 5–500 Hz with a total of 888 data-points, see Fig. 2c) to avoid training the models where sufficient number of data-points were not available. The neural networks were able to accurately predict droplet diameter and generation rate in both dripping and jetting regimes, as given in Table 1. When comparing against the test-set, the neural networks were able to predict the droplet diameter with a mean absolute error (MAE) of 10 μm and 6 μm for dripping and jetting regimes, respectively. Additionally, the neural networks predicted the generation rate with an MAE of less than 20 Hz and 16 Hz for dripping and jetting regimes, respectively. The mean absolute percentage error (MAPE) for generation rate was observed to be about three times the MAPE for droplet diameter (see Table 1). This can be explained by the interdependence of droplet generation rate and diameter, for a given dispersed phase flow rate. Based on the conservation of mass given in Eq. (1), it can be concluded that generation rate inversely scales with diameter to the power of three, as given in Eq. (2). If droplet diameter is predicted with an error of $\delta$, using the Taylor series expansion the prediction error for $\frac{1}{D^3}$ is approximated by Eq. (3). Therefore, the prediction error for

generation rate ($\delta_f$) can be approximated by Eq. (4).

$$F \sim \frac{1}{D^3} \tag{2}$$

$$\frac{1}{(D+\delta)^3} \approx \frac{1}{D^3} - \frac{3\delta}{D^4} + O(\delta^2) \tag{3}$$

$$\delta_f \approx \frac{1}{D^3} - \frac{1}{(D+\delta)^3} \approx \frac{3\delta}{D^4} - O(\delta^2) \approx \frac{3\delta}{D^4} \tag{4}$$

By dividing Eq. (4) by Eq. (2), it can be concluded that the percentage error for generation rate ($\varepsilon_f$) is approximately equal to three times the percentage error for droplet diameter ($\varepsilon_d$) as given by Eq. (5), provided that the value of $\delta$ is small.

$$\varepsilon_f = \frac{\delta_f}{F} \approx 1 - \frac{D^3}{(D+\delta)^3} \approx \frac{3\delta}{D} \approx 3\varepsilon_d \tag{5}$$

The fact that the percentage errors of the separately trained models for droplet diameter and generation rate are compatible with the conservation of mass principle, demonstrates that the models are representative of the droplet generation phenomenon. It must be noted that this error ratio does not necessarily hold true for all of the data-points, further emphasizing the importance of having two separately trained models for diameter and generation rate to enable accuracy-checking of the predictive models, later used in the design automation stage.

To compare the performance of the developed predictive models to the existing approaches in the literature, the data-points were fed to previously proposed scaling laws for predicting the performance of flow-focusing droplet generators (further introduced in Supplementary Note 4)[16,41]. As demonstrated in Fig. 3a, the scaling laws, although great tools for understanding

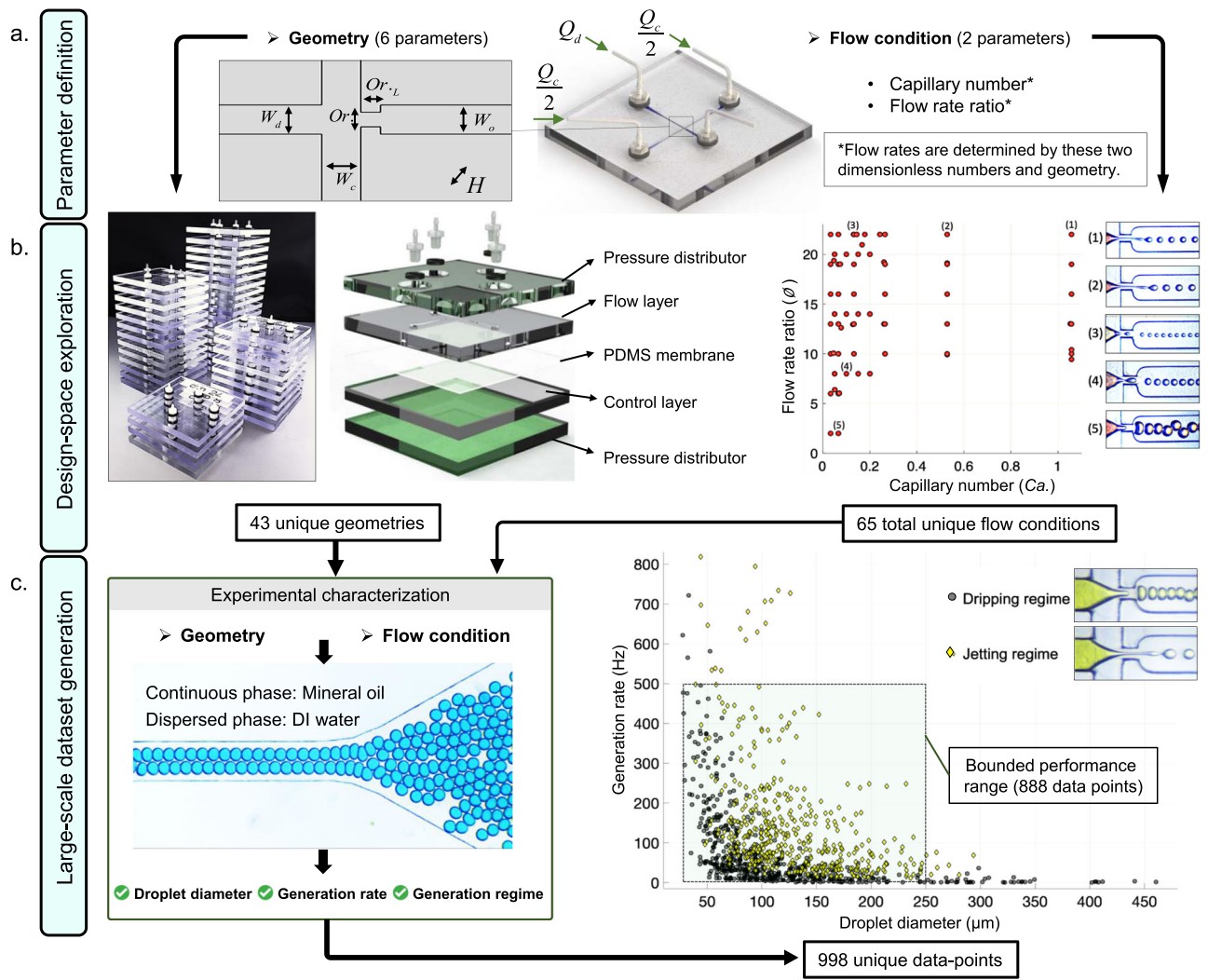

**Fig. 2 Rapid prototyping enables efficient generation of data-sets suitable for training machine learning algorithms. a** The six parameters defining a flow-focusing geometry were varied according to an orthogonal design of experiments. Capillary number and flow rate ratio were used to determine the flow rates for each device. **b** A total of 43 devices were fabricated using a low-cost micromill and assembly technique. Each device is tested over a wide range of flow conditions to generate a large-scale data-set. **c** For each experiment, droplet diameter, generation rate, and generation regime were recorded. A total of 998 data-points were collected of the observed performance and their associated design parameters. The machine learning algorithms were trained on a bounded performance range of 25–250 μm for droplet diameter and 5–500 Hz for generation rate. The bounded range included 888 data-points with a comparable representation of both dripping and jetting regimes.

**Table 1 Neural networks enable accurate performance prediction in flow-focusing droplet generation. The metrics are reported for a 20% test-set (training the models on 80% of the data and leaving 20% for the test-set).**

| Parameter | Regime | $R^2$* | RMSE** | MAPE*** | MAE**** |
|---|---|---|---|---|---|
| Droplet diameter | Dripping | 0.893 ± 0.029 | 13.1 ± 1.6 μm | 11.2 ± 1.3% | 9.9 ± 1.2 μm |
| Droplet diameter | Jetting | 0.966 ± 0.010 | 8.2 ± 1.3 μm | 4.8 ± 0.5% | 5.9 ± 0.8 μm |
| Generation rate | Dripping | 0.889 ± 0.026 | 31.7 ± 5.8 Hz | 33.5 ± 4.2% | 19.6 ± 2.7 Hz |
| Generation rate | Jetting | 0.956 ± 0.009 | 21.9 ± 2.8 Hz | 15.8 ± 2.9% | 15.4 ± 2.1 Hz |

*$R^2$ coefficient of determination, **Root mean square error, ***Mean absolute percentage error, ****Mean absolute error. The provided values are reported using the average plus-minus (±) the standard deviation for ten different training and testing sessions. For each session the test-set and train-set were randomly chosen.

the overall effect of parameters on performance, fall short on accurate performance prediction in comparison to machine learning based models (Fig. 3b).

To demonstrate the efficacy of the developed tool in predicting the performance of new design parameters on which they were not trained (i.e., unseen design parameters), six unseen flow conditions (see Supplementary Note 5) were fed to DAFD and the

accuracy of performance prediction was evaluated by running these six new experiments. DAFD predicted the generation regime with 100% accuracy, and showed an MAE (MAPE) of 5.41 μm (7.01%) and 38.1 Hz (24.2%) in predicting diameter and generation rate, respectively (Fig. 3c).

Finally, the effect of data-set size on the accuracy of the predictive neural networks was studied through a data reduction

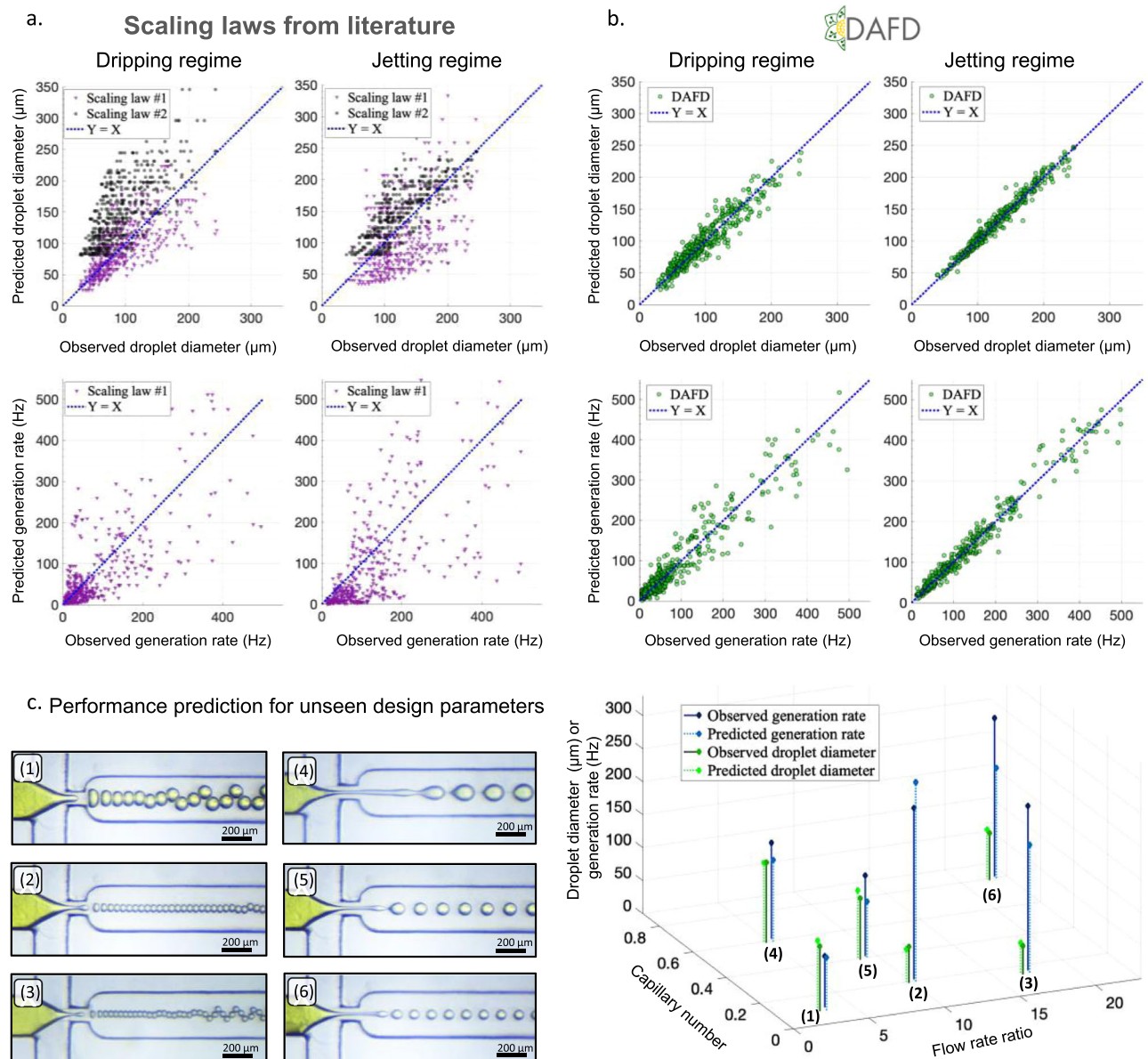

**Fig. 3 Neural networks provide superior accuracy in predicting the performance of microfluidic flow-focusing devices. a** Scaling laws, despite being valuable tools for understanding the dynamics of droplet generation, fall short on accurate performance prediction. Scaling law #1[16] & #2[41] performed similarly in predicting droplet diameter. For generation rate prediction, scaling law #1 was not accurate, and scaling law #2 could not be used for performance prediction. **b** The developed neural networks predicted the droplet diameter and generation rate in both dripping and jetting regimes accurately. Droplet diameter prediction was more accurate in comparison to predicting generation rate, and performance prediction in jetting regime was more accurate in comparison to dripping regime (see Table 1). The predicted performance are depicted for 20-fold cross-validation. **c** The neural networks showed an MAPE of 7% for droplet diameter and less than 25% for generation rate while predicting the performance of unseen design parameters, which is similar to the accuracy observed for previously seen design parameters.

study. By training the models on increasingly larger sub-samples (starting from 50 data-points up to 325 data-points for each regime) and testing against a 20% randomly selected sub-sample of the original data-set. It can be concluded that approximately 250–300 informative data-points for the dripping regime and 200–250 informative data-points for the jetting regime (a total of approximately 500 data-points) would yield a relatively similar accuracy to the full bounded data-set with an 80% train-set ($0.8 \cdot 888 \approx 710$ data-points), as shown in Fig. 4.

**Generalizable performance prediction**. In our original data-set, variations of geometry and flow condition were thoroughly

considered. However, fluid properties were kept constant and DI water and NF 350 mineral oil were used to produce droplets. Depending on the application, a different fluid combination with different fluid properties (viscosity, surface tension, etc.) can be used. To enable performance prediction of droplet generation with different fluid combinations, two different workflows were established. First, an automated data-to-model machine learning framework, DAFD Neural Optimizer, was developed to train neural networks from scratch. Second, the neural networks trained on the original data-set were utilized as base-models (pre-trained models) for automated transfer learning to allow researchers to fine tune the models for new fluid combinations[42]. Thus, the microfluidic community can build custom predictive

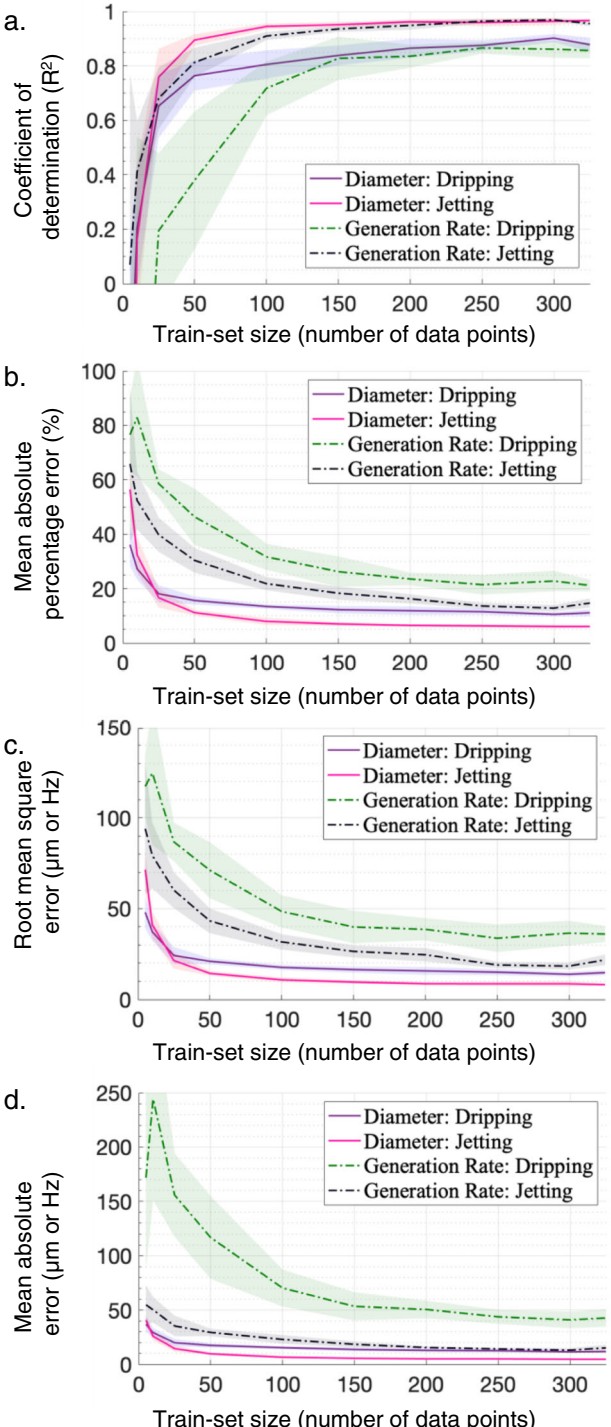

**Fig. 4 Effect of data-set size on the accuracy of performance prediction.**
The data reduction study suggests that a relatively similar performance prediction accuracy can be achieved with fewer data-points. Based on the four criteria of **a** coefficient of determination, **b** mean absolute percentage error, **c** root mean square error, and **d** mean absolute error it can be concluded that approximately 250–300 informative data-points for the dripping regime and 200–250 informative data-points for the jetting regime would yield a relatively similar accuracy. The train-set (up to 80%) and test-set (20%) were randomly selected ($N = 10$) from the existing 888 data-points.

models using DAFD Neural Optimizer (for large-scale data-sets) or transfer learning (for small-scale data-sets), as demonstrated in Fig. 5.

A web-based tool was developed that enables researchers to upload a custom tabular data-set and generate optimized neural networks, without the need for machine learning expertise. This tool utilizes an automated machine learning approach to design, train, and evaluate neural networks in an easy to use interface. DAFD Neural Optimizer can search for optimal hyperparameters of neural networks while allowing for data-normalization method, test-set size, validation method, number of folds, and the evaluation metric to be specified (see Supplementary Figs. 6 and 7). To demonstrate the efficacy of this tool, the previously generated large-scale data-set was uploaded to Neural Optimizer and the trained models were compared to the neural networks we previously built with expertise and trial and error, and a comparable accuracy in predicting droplet diameter and generation rate was observed (see Fig. 5a). This tool is further explained in Supplementary Note 6.

Training an accurate neural network model requires a relatively large data-set. Consequently, an immediate drawback of training neural network models on small-scale data-sets is over-fitting to the train-set, resulting in models that poorly generalize to the test-set[43]. Transfer learning was shown to be explicitly beneficial in improving the performance of neural network models for small-scale data-sets when another accurate model trained on a relatively similar system, also known as a pre-trained model, is available[44]. To fine tune the pre-trained model to fit a new data-set, the structure and weights of the first few layers (that carry more generic features of the system) of the neural networks are kept unchanged, while the last layers (that carry more specific features) are allowed to be updated[45].

Transfer learning can also be implemented in performance prediction of droplet microfluidics since the high-level dynamics remain the same regardless of the fluid combinations used to generate droplets[46]. Therefore, for a small-scale data-set on droplet generation with a new fluid combination, the models trained on the original data-set (pre-trained models) can be fine tuned to accurately predict the performance, allowing for significantly fewer number of data-points.

To demonstrate the applicability of transfer learning in performance prediction of droplet generators, we generated two new small-scale data-sets, in which we either changed the dispersed phase fluid or the continuous phase fluid. First, lysogeny broth (LB) bacterial cell media (instead of DI water) and NF 350 mineral oil were used to generate droplets and create a small-scale data-set of 36 data-points, with data-points in both generation regimes. Second, light mineral oil with a viscosity of 21.4 mPa s with 2% span 80 as surfactant (instead of NF 350 mineral oil with a viscosity of 57.2 mPa s with 5% span 80) and DI water were used to create a small-scale data-set of 18 data-points in the dripping regime. Training neural networks from scratch on these data-sets resulted in non-generalizable models that over-fitted to the train-set, and performed poorly on the test-set, as shown in Fig. 5b, c. Conversely, by fine tuning the pre-trained models using transfer learning, the performance prediction accuracy on the test-set improved significantly (see Fig. 5b, c). Therefore, by using transfer learning only a fraction of initial data-points is required to achieve a comparable accuracy for a new fluid combination. Additionally, transfer learning with 18 data-points per regime showed higher accuracy for the LB bacterial media data-set in comparison to the light mineral oil data-set. This can be attributed to the smaller

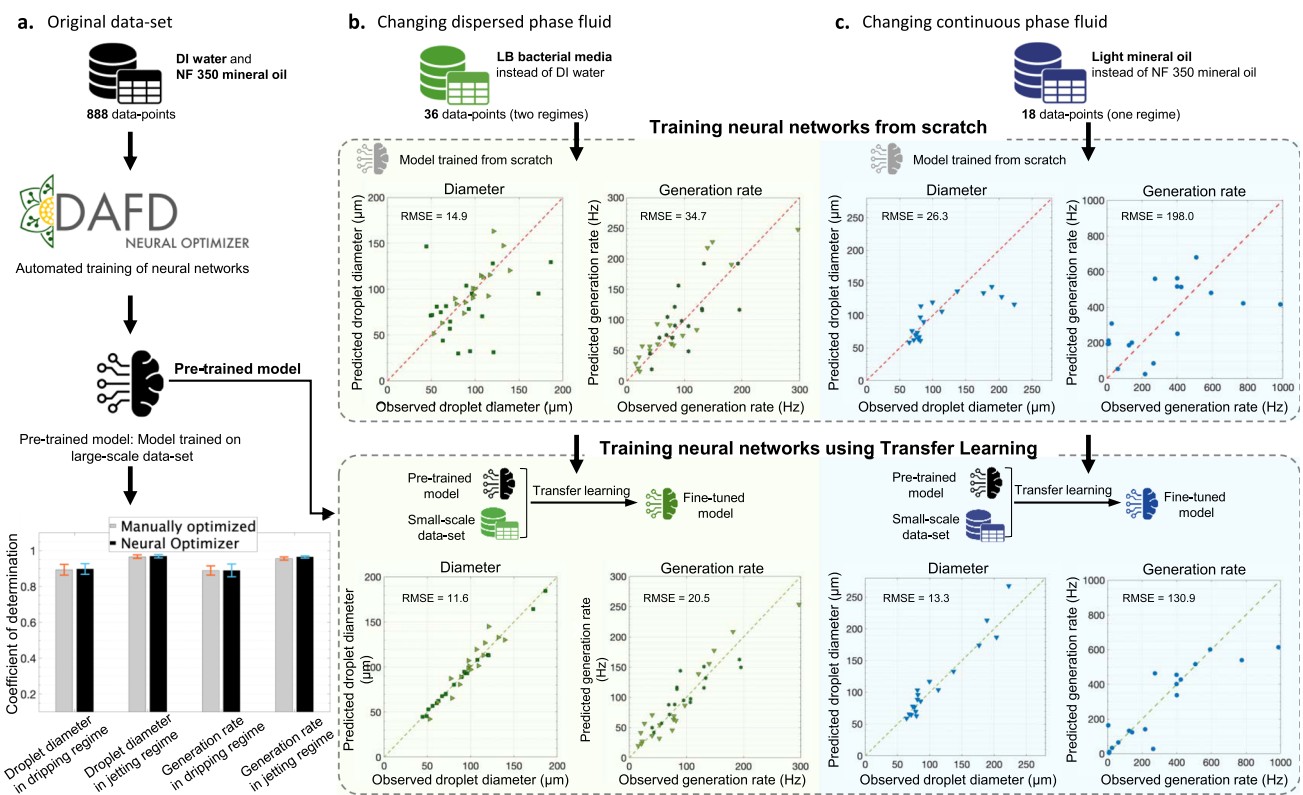

**Fig. 5 Generalizing performance prediction to new fluid combinations. a** Large-scale data-sets on any fluid combinations can be uploaded to DAFD Neural Optimizer to generate optimized neural networks without requiring extensive machine learning expertise, with comparable accuracy to models manually optimized. Error bars represent two standard deviations calculated based on ten different training and testing sessions where the test-set and train-set were randomly chosen. **b** For a small-scale data-set on LB bacterial cell media (instead of DI water) and NF 350 mineral oil, training neural networks from scratch resulted in inaccurate models. However, transfer learning significantly improved the accuracy of models on the same data-set. **c** For a small-scale data-set on DI water and light mineral oil (instead of NF 350 mineral oil), training neural networks from scratch resulted in inaccurate models. Using transfer learning the accuracy of the trained neural networks was improved significantly, despite the large difference in continuous phase fluid properties. Therefore, the pre-trained models on DI water and NF 350 mineral oil and transfer learning can be used to extend DAFD without requiring large-scale data-sets.

difference in fluid properties between LB bacterial media and DI water, in comparison to light mineral oil and NF 350 mineral oil. Therefore, for a new fluid combination that differs significantly in fluid properties (in comparison to NF 350 mineral oil and DI water) more data-points are required for an accurate transfer learning. Researchers can use the information provided in Table 1, and compare the performance of their neural networks trained on a data-set (that conforms with the data-set generated in this study in terms of parameter normalization and placement) of new fluid combinations to the performance of the predictive models developed in this study to determine if sufficient number of data-points are collected.

**Design automation**. The developed design automation tool converts the user-specified performance to the geometry and flow rates required to achieve that desired performance. Droplet diameter (25–250 µm), generation rate (5–500 Hz), and optional design constraints can be specified in the tool. Design automation is achieved by finding a point on the data-set closest to the desired performance, adjusting the design parameters, evaluating a cost function using the predictive models, and minimizing it until the desired performance is reached, as described in Supplementary Note 8.

As a first step to verify the accuracy of design automation, eight different droplet diameters ranging from 25 to 200 µm were specified as the desired performance. The proposed designs (see

Supplementary Note 9) were fabricated and the flow rates were set to the values given by the tool. For the specified droplet diameters an MAE (MAPE) of 4.3 µm (5.0%) was achieved between the specified and observed droplet diameters. A maximum diameter deviation of 12.3 µm (for 200 µm droplets) and a maximum percentage error of 16% (for 25 µm droplets) was observed (see Fig. 6a).

In several applications, accurate control over the droplet generation rate as well as the diameter is essential. To demonstrate the capability of DAFD to design droplet generators that deliver a user-specified performance, a variety of droplet diameter and generation rate combinations were specified. Three droplet diameters and seven different generation rates were specified and the proposed designs (see Supplementary Note 9) were tested, and the observed and desired performance were compared. For a droplet diameter of 100 µm, an MAE (MAPE) of 4.44 µm (4.4%) and 33 Hz (12.2%) was observed for droplet diameter and generation rate, respectively (see Fig. 6b). These values were observed to be 4.36 µm (5.8%) and 33 Hz (14.4%) and 3.92 µm (7.8%) and 41.3 Hz (38.8%) for droplet diameters of 75 µm and 50 µm, respectively (see Fig. 6c, d). The highest error of 14.8 µm and 101 Hz was observed for a droplet diameter of 50 µm and a generation rate of 50 Hz. This can be attributed to the extremely low flow rates (0.206 µl/min DI water and 0.272 ml/h mineral oil) required for this performance, which are prone to experimental errors (expansions, contractions, and movements in syringes and tubing and syringe pump accuracy). Excluding this extreme data-

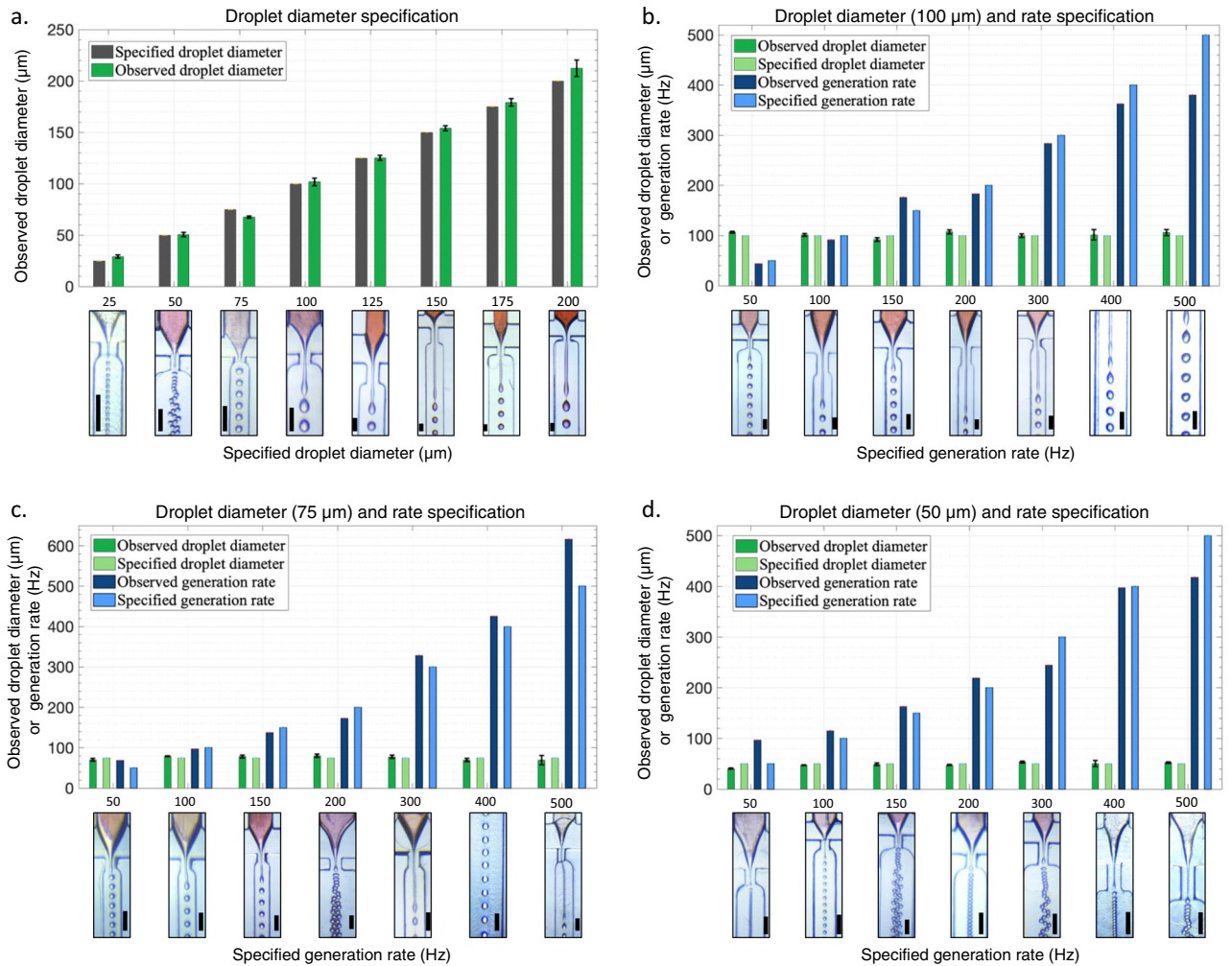

**Fig. 6 Machine learning based design automation of microfluidic flow-focusing droplet generators. a** While specifying only droplet diameter, an MAE (MAPE) of 4.2 μm (5.0%) between the specified and observed diameters was observed. **b** When specifying both a droplet diameter of 100 μm and generation rates ranging from 50 to 500 Hz, an MAE (MAPE) of 4.44 μm (4.4%) and 33 Hz (12.2%) were observed. **c** For specifying both a diameter of 75 μm and generation rate, an MAE (MAPE) of 4.36 μm (5.8%) for droplet diameter and 33 Hz (14.4%) for generation rate were observed. **d** For specifying generation rate and a droplet diameter of 50 μm an MAE (MAPE) of 3.92 μm (7.8%) and 41.3 Hz (38.8%) were observed. The scale bars represent 200 μm. Error bars represent two standard deviations and are calculated by analyzing the variations in droplet diameter (droplet polydispersity) for each experiment.

point, the developed design automation tool delivers a user-specified performance with an MAE (MAPE) of 3.7 μm (4.2%) and 32.5 Hz (11.5%) for droplet diameter and generation rate, respectively. These minor deviations can be easily adjusted by changing the flow rates of the dispersed and continuous phases to reach the desired performance as discussed in the design tolerance prediction section. It must be noted that the design automation accuracy exceeded the accuracy of the predictive models because of the accuracy-checking (redundancy) in the predictive models that enabled the introduction of the inferred droplet diameter (see "Methods: Design automation" section).

**Design tolerance prediction**. The experimentally observed performance of a droplet generator suggested by the design automation tool can be affected by the tolerances in fabrication and syringe pumps. To this end, the developed predictive models were used to quantify the effect of these tolerances on the performance of a given design. Based on the droplet generator design and a user-specified tolerance, three different values for each design parameter were considered (in the range of the designed value plus/minus the tolerance). Using quasi-Monte

Carlo sampling[47] and the neural network based predictive models, principal parameters affecting droplet diameter and generation rate were identified using variance-based sensitivity analysis in an automated manner (see "Methods: Design tolerance study" section).

Once the principal parameters affecting droplet diameter and generation rate are identified, the effect of the tolerance for the remaining parameters was plotted against the principal parameter, as shown in Fig. 7c, d. The principal parameters could be different for droplet diameter and generation rate, and it could vary depending on the design of the droplet generator, suggesting the importance of the developed design-specific tolerance prediction tool. The flow rates of the continuous and dispersed phases can be adjusted to account for these tolerances. Therefore, the machine learning based predictive models were automated to predict the performance of given a design for flow rates lower and higher than the designed values (two times the user-specified tolerance). As a result, two plots are generated depicting the effect of flow rates on the droplet diameter and generation rate for a given design to guide researchers to adjust flow rates according to the observed performance deviation caused by the tolerances (see Fig. 7b).

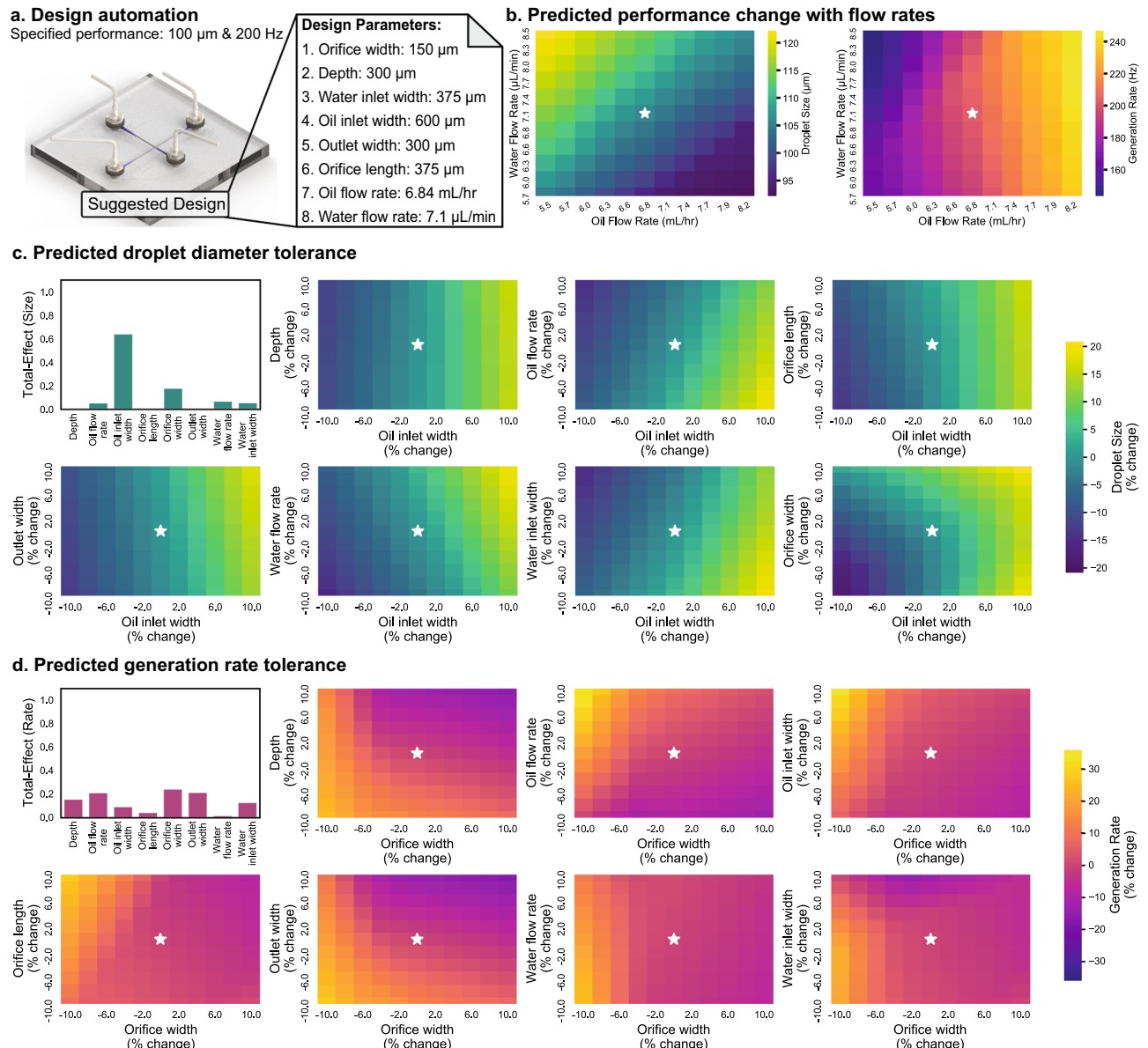

**Fig. 7 Prediction of performance deviations caused by fabrication and flow rate tolerances. a** Once a design is suggested by the tool based on the user-specified desired performance, **b** the variations in droplet diameter and generation rate for the suggested design for changing water and oil flow rates are predicted using the developed neural networks. This helps users adjust for possible performance deviations caused by tolerances in fabrication. **c** Based on the user-specified tolerance, by using the developed predictive models, parameter sensitivities are quantified through Sobol sensitivity analysis. This allows the users to identify the most influential design parameters for a given design that can cause the most amount of performance deviations for a similar percentage tolerance. The diameter changes caused by tolerances are predicted and plotted with the most effective parameter as the *x*-axis and the remaining parameters on the *y*-axis. **d** The same process explained in **c** is used to identify generation rate sensitivity to tolerances in design parameters and plot the changes in generation rate caused by fabrication tolerances.

**Case-study: single-cell encapsulation**. The ability to measure the properties of single cells isolated from a large population is one of the main motivations of droplet microfluidics[48–50]. With few exceptions, cell encapsulation occurs through a random process that is dictated by the Poisson distribution[51]. To demonstrate that performance prediction of droplet generators enables further design automation capabilities (e.g., single-cell encapsulation, droplet merging, and performance-driven design optimization), cell concentration calculation for single-cell encapsulation was incorporated into the design automation tool (see "Methods" section).

To examine the accuracy of the developed tool in providing the required cell concentration to ensure single-cell encapsulation for a user-specified performance, a droplet diameter of 50 µm, a

generation rate of 150 Hz, and a ratio of cells to droplets of 0.05 was specified. Additionally, two design constraints were specified in the software to test its capability in delivering the desired performance while imposing design constraints. First, the lowest aspect ratio allowable in the tool (value of 1) was specified as a design constraint to keep the device shallow and maintain the cells in the plane of focus. Second, the lowest normalized water inlet allowable in the tool (value of 2) was specified as a design constraint to avoid secondary flows inside the inlet channel that could trap the cells.

Naturally, because of the specified design constraints, the geometry suggested by DAFD was different in comparison to the geometry suggested for a diameter of 50 µm, a generation rate of

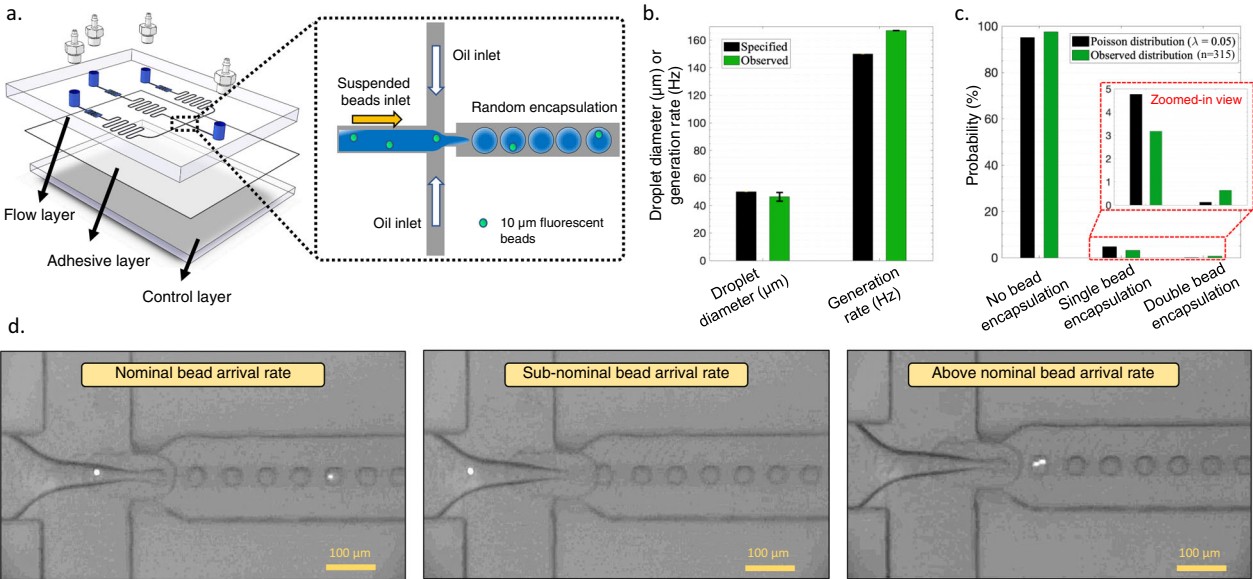

**Fig. 8 Single-cell encapsulation with user-specified performance. a** The desired performance of 50 μm and 150 Hz was specified and the design suggested by the tool was fabricated. Polystyrene beads 10 μm in size were used as cell surrogates and were suspended in DI water at a concentration given by the tool (922.9 beads/μl). **b** By setting the flow rates to the suggested values, a droplet diameter of 46.3 μm at 167 Hz was achieved. **c** Within a total of 315 continuous events, 10 events of single-bead encapsulation, and 2 events of double-bead encapsulation were observed, which closely follows the Poisson distribution. **d** Snapshots of the experiment further emphasize the random but probabilistic nature of single-cell encapsulation. Coloring was adjusted for better visualization.

150 Hz in the design automation section (without design constraints). The suggested device was fabricated and tested with the provided flow rates and cell concentration while using 10 μm fluorescent beads as cell surrogates. A droplet diameter of 46.3 μm and a generation rate of 167 Hz was observed (Fig. 8b), demonstrating the efficacy of the design automation tool in proposing designs with user-specified constraints. Additionally, the bead encapsulation rate followed the Poisson distribution closely (Fig. 8c). Slightly higher than expected double-bead encapsulation events were observed, which can be attributed to the weak hydrophobic surface properties of polystyrene that facilitate aggregations of beads suspended in DI water.

## Discussion

We presented a web-based open-source tool that leverages machine learning to accurately predict the performance of flow-focusing droplet generators. This enabled us to build a design automation tool that takes user-specified performance as input and provides a geometry and flow condition to achieve the desired performance in return, thus, eliminating the need for resource-intensive design iterations.

The current version of the developed tool supports droplet generation with DI water and mineral oil. We demonstrated that this tool can be further extended to support additional fluid combinations either by using Neural Optimizer (for large-scale data-sets) or through transfer learning (for small-scale data-sets). We believe that the automated scheme of Neural Optimizer and the efficacy of transfer learning in reducing the number of required data-points would enable the community to develop custom predictive models for a wide variety of fluid combinations, without requiring extensive machine learning expertise or vast experimental resources. Furthermore, the open-source nature of the tool allows for these custom predictive models to be easily integrated into the developed workflow. Therefore, we envision future work leading to more comprehensive design automation tools for droplet microfluidics developed by the community to support a diverse range of applications.

The framework established in this study is application-agnostic, and the authors believe that a similar workflow can be implemented in other microfluidic components such as micro-mixers[52] or even non-microfluidic components such 3D printed lattices[53] to achieve design automation of complex phenomena. Therefore, through design of experiments and rapid-prototyping, structured data-sets (conforming to a tabular format) can be generated and loaded to DAFD Neural Optimizer to train optimal predictive neural networks. Using these predictive models and by implementing iterative search algorithms such as the one used in this study, researchers can achieve accurate design automation.

We also demonstrated that the developed tool can be used to quantify and account for performance deviations caused by possible tolerances in fabrication or testing. Additionally, the ability to automatically design droplet generators that deliver the desired performance enabled automated calculation of cell concentration for single-cell encapsulation, as demonstrated in this study. Our tool can be extended by the community to accommodate further design automation capabilities. For instance, the developed tool can be integrated with models that predict the path that individual droplets take in a network of microfluidic channels[54], to enable both performance and behavior prediction.

Finally, the developed design automation tool can be integrated with other microfluidic computer-aided design tools to enable more sophisticated design automation[55–63]. We envision, our tool's parameter discovery and refinement capabilities to be augmented with high-level CAD software being developed. The high-level software can translate liquid interactions (merging, splitting, branching, etc.) into a directed graph data structure using a high-level programming language similar to Verilog[64]. That language can be compiled into a parameter-free version of a netlist file called MINT[65]. MINT can be read by emerging software tools including 3DμF[66]. Using this file, 3DμF's design abilities, and DAFD's parameter exploration, a fully realized design can be created quickly without having to have monolithic designs that encode both the structure and function in a non-modifiable manner. Such tools can significantly reduce the barrier

to entry to microfluidics and allow it to play an integral role in numerous fields without demanding substantial microfluidic expertise or resources.

## Methods

**Reagents and materials.** For the original large-scale data-set NF 350 mineral oil with a viscosity of 57.2 mPa s and a specific density of 0.857 was used as the continuous phase. Five percent of V/V Span 80 surfactant (Sigma-Aldrich) was added to the oil to reduce the surface tension for higher generation rates and droplet stability. DI water with added food color for better visualization was used as the dispersed phase. For the small-scale data-sets lysogeny broth (LB) bacterial cell media (L3022, Sigma Aldrich) and light mineral oil with a viscosity of 21.4 mPa s (M8410, Sigma Aldrich) were used for droplet generation. Microfluidic devices were milled from polycarbonate sheets with a thickness of 5.56 mm (McMaster-Carr). Slygard 184 Silicone elastomer kit (Dow Corning) was used to make thin layers of Polydimethylsiloxane (PDMS) to seal the microfluidic geometry.

**Fabrication and assembly of microfluidic devices.** A low-cost desktop CNC micromill (Othermill/Bantam Tools) was used to mill out the flow and control layers of the microfluidic device on a polycarbonate substrate. A layer of PDMS was placed between the flow and control layer to seal each device. Two layers of pressure distributors were milled out of polycarbonate, one pressure distributor was placed above the flow layer and one was placed beneath the control layer. Each device, with a total of five layers, was clamped to increase the sealing pressure and achieve a uniform seal.

**Droplet generation.** Droplet generation was achieved by flowing the continuous and dispersed phases through a flow-focusing geometry. To dampen the flow rate fluctuations induced by the syringe pumps, 50 cm long flexible PVC tubing (McMaster-Carr) with an inner diameter of 1.6 mm and outer diameter of 3.2 mm was used to introduce the fluids to the microfluidic devices. Both phases were filtered through a 0.45 μm polyvinylidene fluoride (PVDF) filter (Millipore). To ensure cross-lab validity of the results, New-Era syringe pumps were used to generate the data-set and Harvard Apparatus syringe pumps were used to verify the accuracy of DAFD.

**Image acquisition and processing.** A high-speed camera (IDT Xstream) mounted on a stereo-microscope (AmScope) was used to record experiments with frame rates up to 18,000 frames per second depending on the speed of the experiments. An 18,000 Lumen LED light source (Expert Digital Imaging) was placed underneath the microfluidic device to ensure sufficient illumination. Each video was analyzed using an open-source image-processing software we previously developed, uDROP, for microfluidic flow-focusing droplet generation. This software records generation rate, droplet diameter, and polydispersity of droplets and is available at: https://github.com/CIDARLAB/uDrop-Generation.

**Flow-focusing geometry definition.** The six parameters defining a flow-focusing geometry were all normalized to the value of the orifice width (except orifice width itself). The values for these geometric parameters were varied according to the values observed in the literature while considering the fabrication limits of desktop micromilling (minimum features size of 75 μm), as further explained in Supplementary Note 2.

**Flow condition definition.** Capillary number and flow rate ratio are the two main dimensionless numbers that can define the flow condition in a flow-focusing geometry[20]. Given the geometry and these two dimensionless number the flow rates of continuous and dispersed phases can be calculated using Eq. (6)[23]:

$$Q_c = \frac{Ca. \cdot \sigma \cdot H \cdot W_c}{\mu_c W_d \left[ \frac{1}{Or.} - \frac{1}{2W_c} \right]}$$
$$Q_d = \frac{Q_c}{\phi},$$
(6)

where $Ca.$ is capillary number, $\phi$ is flow rate ratio, $\mu_c$ represents dynamic viscosity, $Q_c$ is oil flow rate, $\sigma$ denotes surface tension between the continuous and dispersed phases, $H$ is channel depth, and $W_d$, $W_c$, and $Or.$ are water inlet, oil inlet, and orifice widths, respectively. In this study, capillary number and flow rate ratio were varied according to the values observed in the literature and were used to calculate the oil and water flow rates for the designed flow-focusing geometries, as further explained in Supplementary Note 2.

**Performance prediction.** Multi-layer feed-forward neural networks were used for performance prediction. The input dimensions of the neural networks were the eight design parameters of flow-focusing droplet generators and the output layer was a single node representing the predicted droplet generation regime, diameter, or rate. Depending on the complexity of the prediction task, three or four hidden layers with rectified linear activation functions were included in the network

structure to achieve an acceptable performance prediction. Adam optimizer was used to minimize the classification or regression cost. Cross-validation, dropout regularization, and early stopping were used during the training process to avoid over-fitting to the train-set. This enabled the models to be generalized from the train-set to test-set. The classifier (neural network model for predicting droplet generation regime) was developed using all 998 points of the dataset. The regression models (neural networks for predicting the droplet diameter and generation rate) were developed using the bounded performance range. This range with a total of 888 data-points enabled a higher accuracy for the predictive models by avoiding training them on design spaces without sufficient number of data-points, as shown in Fig. 2c. To train both classification and regression models, the data-set was split into train and test sets with an 80–20% ratio, respectively. Neural networks were trained on the train-set, and their performance was evaluated using test-set (which was hidden from the model during the training stage) to provide an unbiased performance evaluation of the developed models.

Python built-in data analysis and machine learning packages (Numpy, Pandas, and Scikit-learn) were used for statistical analysis, data normalization, train-test split sampling, and reporting the performance metrics (accuracy, MSE, etc.) of models. The neural networks were implemented using Keras package. Additional information on the neural networks is provided in the Supplementary Note 3. The source-code and instructions on running the neural network models are available on GitHub at: https://github.com/CIDARLAB/DAFD.

**Data reduction study.** The bounded dataset (888 data-points) consists of 474 data-points in the dripping regime and 414 data-points in the jetting regime, therefore, the size of the randomly selected sub-samples of the train-sets were incremented as 50, 100, 150, 200, 250, 300, and 325 data-points (the maximum train-set size was set to be smaller than the train-set size for jetting regime, i.e., $0.8 \times 414 = 331$ data-points). These sub-samples were then used to train all four DAFD neural networks and evaluated on the test-set with metrics previously described. The test-set was randomly selected from the bounded dataset (20%). This process was repeated ten times to account for possible biases in random selection.

**Neural optimizer.** The software back-end was based on Python, using Flask as the microweb framework, Scikit-learn, and Keras packages for machine learning and neural network implementation. The front-end was built with Bootstrap and jQuery. Neural Optimizer provides a GUI-based service to build and deploy neural networks at the click of buttons and keeps algorithm implementation, data pipe-line, and codes hidden from the view. The pipeline for building machine learning is parameterized and fed to various custom-built Python functions (e.g., data parser, neural network architecture builder, performance metrics, and evaluation methods selectors, etc.). The software is built following the Model-View-Controller framework[67], where all data input and requests are handled by the view and passed through several routers to the controller, which execute the corresponding algorithms and call the model to interpret the neural network architecture. The machine learning pipeline is implemented via the Pipeline module of Scikit-learn library, which makes the modular implementation of cross-validation, train-holdout validation, and hyperparameter optimization approaches possible. More details on the algorithms used to develop Neural Optimizer are provided in the Supplementary Note 6. Neural Optimizer source code is available at: https://github.com/CIDARLAB/neural-optimizer.

**Transfer learning.** Two small-scale data-sets on droplet generation using six different flow-focusing geometries were generated. The first data-set was generated using NF 350 mineral oil and lysogeny broth (LB) bacterial cell media (L3022, Sigma Aldrich), for a total of 36 unique data-points with 19 and 17 data-points in dripping and jetting regimes, respectively. The second data-set was generated using DI water and light mineral oil with a viscosity of 21.4 mPa s (M8410, Sigma Aldrich) with added 2% Span 80 (instead of NF 350 mineral oil with a viscosity of 57.2 mPa s, with 5% added Span 80), for a total 18 data-points in the dripping regime. The surface tension difference between DI water and mineral oil, DI water and light mineral oil, and LB bacterial cell media and mineral oil was assumed to be negligible. For the LB bacterial cell media, the dataset was split into four non-overlapping folds. To compare the performance of transfer learning to a neural network trained from scratch, both neural network models (with and without transfer learning) were trained on the same 3-folds (train-set) and their performance was evaluated against the 4th fold (test-set). This process was repeated by training new models on the other 3-folds sets until all folds are used as the test-set once. To implement transfer learning, the structure and the weights of the optimized neural network models trained on the original dataset (888 data-points on DI water and mineral oil) were saved (called pre-trained models). The first two layers of the pre-trained models were loaded without the ability to be re-trained (frozen layers) and the weights of the last two layers of the new models were updated to better fit the new dataset. A similar approach was used for the light mineral oil dataset, with the slight change that only the first layer of the pre-trained model was frozen, due to the greater fluid properties differences between this dataset and the original dataset, and additional layers were added to better fit the light mineral oil dataset. Data normalization and performance evaluation are done as previously described in the performance prediction methods. Keras package is

used to save and load the pre-trained models and fine tune the new models. Transfer learning source code is available at: https://github.com/CIDARLAB/neural-optimizer.

**Design automation.** The design automation algorithm starts with finding a point in the dataset, which has an observed performance closest to the user-specified desired performance. To this end, all points in the dataset are ranked according to a fitness value based on the specified desired performance and design constraints (see Supplementary Note 8). If a data-point exists with an observed and predicted performance close to the desired performance (within the tolerance of the predictive models, see Supplementary Note 8), that point on the dataset is returned as the final design. However, if the performance difference between the data-point and the desired performance is outside the tolerance of the predictive models, then that point is taken as the starting iteration for an optimization process to achieve the desired performance. The optimization algorithm adjusts the design parameters by increasing and decreasing each parameter by a same normalized step-size. Given that there are 8 design parameters in total, 16 new devices are proposed at each iteration. The predictive models are then used to predict the performance of these 16 new designs. A cost function is evaluated for the proposed designs to determine the optimal design parameters at each iteration, as given in Eq. (7):

$$C(x) = \left| D_{\text{des.}} - D_{\text{pred.}} \right| + \left| F_{\text{des.}} - F_{\text{pred.}} \right| + \left| D_{\text{des.}} - D_{\text{inf.}} \right|, \tag{7}$$

where $D_{\text{des.}}$ is the desired droplet diameter, $D_{\text{pred.}}$ is the predicted droplet diameter, $F_{\text{des.}}$ is the desired generation rate, $F_{\text{pred.}}$ is the predicted generation rate, and $D_{\text{inf.}}$ is the inferred droplet diameter (i.e., using the droplet volume, which is calculated by dividing the dispersed phase flow rate by the predicted generation rate) calculated by Eq. (8):

$$D_{\text{inf.}} = 10^{6.3} \sqrt{\frac{Q_w}{F_{\text{pred.}}} \cdot \frac{6}{\pi}}, \tag{8}$$

where $D_{\text{inf.}}$ is the inferred droplet diameter in μm and $Q_w$ is the dispersed phase flow rate in $m^3/s$. $D_{\text{inf.}}$ is added to the cost function to ensure the compliance of the neural networks that predict the droplet diameter and generation rate. This avoids areas in the design space where either one or both models predict the performance less accurately. Thus, allowing the design automation accuracy to surpass the accuracy of the predictive models in most cases. Once the cost function of the new designs are calculated, the design that reduced the cost function the most is taken as the new starting design for the next iteration, until accurate design automation is achieved. The algorithms are implemented in Python and NumPy library was used to speed up our computations. This algorithm is further explained in Supplementary Note 8. The algorithms used for design automation were created by our group and the source-code is available on GitHub: https://github.com/CIDARLAB/DAFD.

**Design tolerance study.** The relative effect of each design feature on device performance was first evaluated with variance-based sensitivity analysis[68,69]. In brief, quasi Monte-Carlo samples are generated within the bounds of parameter hypercube set by a user-input tolerance. Next, the variances from first and second order interactions are evaluated as:

$$\text{Var}(X_i) = \text{Var}_{X_i}(E[Y|X_i]), \tag{9}$$

$$\text{Var}(X_{ij}) = \text{Var}_{X_{ij}}(E[Y|X_i, X_j]) - \text{Var}(X_i) - \text{Var}(X_j), \tag{10}$$

where $X$ represents input features and $Y$ is the output performance (in this case droplet diameter or generation rate). Once variances from first and second order interactions are calculated, the total-effect index of each parameter is found, which represents the effect of a single parameter on total variance through both first and second order interactions. The design feature with the highest total-effect index value was then perturbed in a "tolerance grid" along with every other design feature. Each combination was then run through the machine learning based predictive models and visualized in a heatmap to show how changes to each feature affect the performance. The effects of continuous and dispersed flow rates was also predicted to provide a guideline for microfluidic operators to understand how to correct possible deviations in performance caused by tolerances in fabrication or testing. The tolerance study is an available option for both performance prediction and design automation for all users at http://dafdcad.org. Variance-based sensitivity analysis was integrated into the tool using the SALib library[70]. The source-code for tolerance study is available on https://github.com/CIDARLAB/DAFD.

**Cell concentration calculation.** Cell encapsulation often occurs through a random process dictated by Poisson distribution[51], as given by Eq. (11):

$$P(\lambda, k) = e^{-\lambda} \frac{\lambda^k}{k!}, \tag{11}$$

where $\lambda$ is the average number of cells per droplet volume and $k$ is the number of cells encapsulated in a droplet. $P(\lambda, k)$ is the probability that $k$ cells are

encapsulated in a single droplet for a given cell concentration ($\lambda$). To ensure single-cell encapsulation, the cells entering the device are out-numbered by the number of droplets generated per unit of time, therefore, $\lambda$ is typically kept between 0.05–0.1[51]. The ability of the developed tool to predict the dispersed phase flow rate and droplet generation rate enables the required cell concentration for single-cell encapsulation to be calculated using Eq. (12):

$$C_{\text{cells}} = 60 \cdot \frac{F \cdot \lambda}{Q_w}, \tag{12}$$

where $C_{\text{cells}}$ is the inlet cell concentration (cells/μl), $F$ is droplet generation rate (Hz), and $Q_w$ is dispersed phase flow rate (μl/min).

**Single-cell (bead) encapsulation.** Ten micrometer yellow-green fluorescent (505/515 nm) polystyrene microspheres (Thermo Fisher) were used as cell surrogates. A droplet diameter of 50 μm at 150 Hz was specified to the design automation tool while constraining the design to have the lowest allowable aspect ratio (the ratio of channel depth to orifice width), and normalized water inlet width to keep the beads at the plane of focus and prevent them from getting trapped in a local flow field. The microfluidic device proposed by the tool was milled out of polycarbonate and sealed with a 125 μm thick acrylic adhesive instead of using a PDMS membrane and pressure distributors, to help visualize the experiment with an inverted microscope. The DAFD provided flow condition of a Ca number of 0.132, a flow rate ratio of 10 (0.312 ml/h oil flow rate and 0.52 μl/min water flow rate), and a bead concentration of 922.8 beads/μl suspended in 0.5 molar $CaCl_2$ DI water (to delay bead settling in the syringe and tubing) were used to run the experiments. A high-speed camera (IDT Xstream) mounted on an inverted microscope (Zeiss Axiovert 200M) operating at both bright-field and fluorescent imaging modes was used to record the experiments at 1200 frames per second.

## Data availability
All experimental data-sets generated and used in this study are available for download at: http://dafdcad.org.

## Code availability
All source code generated and used in this study are available on https://github.com/CIDARLAB/DAFD and https://github.com/CIDARLAB/neural-optimizer.

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

## Acknowledgements

We would like to thank Dr. Calin Belta from Boston University for supporting this study. We are thankful to Radhakrishna Sanka for developing the DAFD-3DµF integration module. This material is based on work supported by U.S. Defense Advanced Research Projects Agency (DARPA) Living Foundries award HR0011-15-C-0084. This work was supported by the Grunebaum Faculty Research Fellowship (J.D.C.), the National Library of Medicine R01LM013154-01 (J.D.C.), and the NSF Living Computing Project Award #1522074.

## Author contributions

A.L. developed the research idea, fabricated devices, designed and carried out experiments, and analyzed data. C.R. built the image processing tool, analyzed videos, and built the design automation platform. N.M. developed, optimized, and verified the neural networks for regime classification, performance prediction, and transfer learning. R.M.

developed Neural Optimizer and DAFD's website. D.M. conducted the data-reduction study and developed and integrated the design tolerance study module to DAFD. C.R., N.M., R.M., and D.M. contributed equally. L.O. helped with single-cell encapsulation experiments. J.C. oversaw and partially funded the project. D.D. helped develop the research idea, enabled the platform to build the design automation tool, oversaw, and funded the project. All authors have read and approved this manuscript.

## Competing interests

The authors declare no competing interests.
