## [Peer Review File · Nature Communications]

Reviewers' comments:

Reviewer #1 (Remarks to the Author):

The authors present a web-based tool that either generates designs for microfluidic droplet generators given diameter and production rate requirements or predicts diameter and production rates from a user-provided design. They do so by taking advantage of rapid prototyping to fabricate and automatically evaluate the performance of a wide set of droplet generator designs, thereby generating a size-able data set for machine learning. Trained models perform very well in dripping or jetting regimes given (1) a candidate design, (2) any combination of operating requirements, and (3) on generalization tests using small data sets of different fluids by leveraging transfer learning. Since their model performed well in a variety of conditions, they are confident others in the microfluidic community can leverage their web-tool for similar systems. The paper is well written and the analysis is sound. Since the authors make their DAFD tool freely available, this work can make waves in the microfluidic community.

Given the strong performance of the model and clear execution pathway of their approach, this systematic method to fabricate and evaluate designs can hopefully be implemented in other disciplines beyond the microfluidic community. Since Nature Communications' audience is so broad, the authors should clearly indicate in the manuscript *how* their web tool (or approach to generate the data, and complementary machine learning models) can be adopted in other areas? Can it extend beyond just droplet generation applications in microfluidics? Or can it benefit non-microfluidic applications where rapid prototyping is possible, e.g. designs of 3D printed lattices?

The authors work is possible due to a large test bed of data they generated. Also, the authors show how transfer learning makes it possible for others to upload distinct and relatively smaller datasets to DAFD to yield similar results. However, it's not clear how much training data is enough to produce reliable results. It is straightforward to show this via an "ablation study" wherein the machine learning models are trained on progressively larger random subsamples of the overall data and tested against. This way, you could argue concretely that 998 data points were a sufficient starter set. Equivalently, future DAFD users could estimate if their results, e.g. 36 data points, are enough. Essentially, it would be useful to let the readers understand what data collection requirements there are when using DAFD for their applications.

Do DAFD users have to label their data using the same uDROP image-processing software you developed? If they use a different high speed camera system than what was used in the manuscript, is it safe to pool their data with yours during transfer learning?

It's great that the DAFD models outperform scaling laws from literature as shown in Figure 3a and 3b. Can the authors compute the same performance metrics in Table 2 for these scaling laws to further aid the comparison?

Reviewer #2 (Remarks to the Author):

The authors present a design automation tool for generating microfluidic droplet generators based on hydrodynamic flow focusing. The authors made the software tool available as public domain. I regard it as a good manuscript and I think this could develop into a very useful tool in the future that deserves publication in one of the major fluidic journals.

Nevertheless, in my view the content of the manuscript is not generic enough and does not have enough novelty and impact to justify publication in Nature communications. Main reasons for that are:

i) All experiments to train the network are done with water and mineral oil and do not consider changes in surface tension and viscosity of the dispersed or continuous phase. This limits the predictions for real world scenarios employing a wide range of surfactants for the dispensed phase. The authors presented a tool (Neural Optimizer) that allow users to train the network with their own data. Nevertheless, this does not

compensate for this weakness in my view .

ii) The approach only covers microfluidic droplet generators based on hydrodynamic flow focusing. Other methods such as droplet generation by step emulsification, droplet generation by T-channel configurations, etc. are not discussed and are not covered in the manuscript.

1 Mandatory remarks

1.1 Title

The claim of the title is too broad and not justified. It should be changed into "Machine Learning Guided Design of Microfluidic Droplet Generators Based on Hydrodynamic Flow Focusing" because other methods than "hydrodynamic flow focusing" (as e.g. droplet generation by step emulsification, T-channels, etc.) are not covered at all in the manuscript.

1.2 Droplet diameter AND droplet frequency as two independent entities

The authors claim, that the model can predict droplet diameter AND droplet frequency and create the impression that those would be two independent entities. As a "droplet diameter" corresponds to a "droplet volume" and "droplet volume" times "droplet frequency" results in the "flow rate" which is controlled from outside in the experiments of the manuscript, those two are NOT independent entities. So if you control flow rate from the outside, and can predict EITHER "droplet volume" OR "droplet frequency", you can easily calculate the other entity.

The authors keep this impression alive during the whole manuscript as they always give two entities "predicted droplet diameter" and "predicted frequency" and compare those with the outcome of other approaches from the literature (e.g. in figure 3). Either they should justify this explicitly or they should change that.

In my view, the fact that both parameters are not independent from each other can also easily be seen in the results. The authors state "DAFD ... showed an average error of 5.41 μm (7.01%) and 38.1 Hz (24.2%)." As the error in the droplet volume scales with the power of 3 on the error in droplet diameter, in my view it is clear from the beginning that an error of $\sim 7\%$ in droplet diameter will result in an error of $\sim 21\%$ in droplet volume and frequency.

The same is true for the statement "Droplet diameter prediction was more accurate in comparison to predicting generation rate ..." in the captions of figure 3. In my view this statement is obvious and absolutely not surprising. The authors should clarify to the reader, that droplet diameter and frequency are coupled entities as long as the flow rate is fixed.

1.3 Comparison with references [15] and [34]

The authors compare their work with reference [15] and [34] in figure 3. Both references are using a geometrical arrangement of the flow focusing structure, which is similar, but not the same as in the manuscript under review. Continuous phase and dispensed phase are fed in parallel in [15] and [34] while in the current manuscript they meet orthogonally. This should be mentioned and the consequences of it should be discussed. I expect this to make a difference especially for larger flow rates (equivalent to larger frequencies). Finally, as a consequence, I think it is not a fair statement to claim that the new model based on machine learning is better than the scaling models in the two references as the authors did.

1.4 Viscosity and surface tension

The authors state "In our generated data-set, variations of geometry and flow condition were thoroughly considered. However, fluid properties were kept constant and DI water and mineral oil were used to produce

droplets". Because all experiments are done only with two dedicated fluids, water and oil, I regard the experimental proof of the machine learning approach quite limited. It would be much more interesting if the authors would be able to generate predictions on a wide range of surface tensions (and viscosities) as those parameters change frequently in real world experiments.

2 Minor remarks

2.1

I don't think that the case study in single cell encapsulation (section 2.4.1) adds a lot value to the manuscript. In my view it's just a calculator for cell density to be fed to the dispensed phase and this is not a big deal.

2.2

I don't think that reference [8] is adequate to justify the statement „Large machine footprints and high overhead costs limit the accessibility of liquid handling robots [8] ...“

2.3

I don't think that reference [14], a paper from 2006, justifies the following statement in year 2020: "... adoption of droplet-based platforms in the life sciences has been an exception rather than the norm [14]." This is especially no true, knowing that digital assays building on microdroplets (Biorad, Stilla, etc.) came up heavily during the last decade. Even the statement there would be a "... lack of predictive understanding [17]" refers to a 10 year old paper and there happened a lot during the last 10 years.

Reviewer #3 (Remarks to the Author):

This paper describes software to aid the design and simulation of droplet microfluidic devices for forming emulsions. Overall the approach is useful and interesting and should be valuable to the field. The video demonstration is quite useful and should aid adoption of the technology. The paper is well written, concise, and clear, and the plots useful. Overall, the manuscript is thus quite strong.

My problem with the manuscript is that I don't think the approach will be that impactful in the field. While indeed generating droplet microfluidic devices is important for research applications, generally speaking the people that build the devices are experts in the design and fabrication. This paper will no doubt aid the design component, but similar devices can already be made with intuition and trial and error. Thus, I do not think the paper will be that impactful to experts.

For non experts, this approach only addresses the design challenge. The other challenges for fabricating and successfully operating the devices are still up to the user and, in my opinion, these constitute equal if not even greater challenges. Thus, I do not think the approach will make it substantially easier for non experts to use such devices.

Thus, while the paper is interesting and well done, I am doubtful of the broad impact it will have.

Dear reviewers,

We would like to start with the sincerest thanks to the reviewers for their time and invaluable feedback. We believe the points made by the reviewers and the editor helped us improve the quality of our paper significantly and tailor it to a broader range of audience. We hope that the revised version of our manuscript will be considered for publication in Nature Communications journal. We have carefully addressed each of the reviewer's concerns as outlined below.

Reviewer #1 (Remarks to the Author)

- The authors present a web-based tool that either generates designs for microfluidic droplet generators given diameter and production rate requirements or predicts diameter and production rates from a user-provided design. They do so by taking advantage of rapid prototyping to fabricate and automatically evaluate the performance of a wide set of droplet generator designs, thereby generating a size-able data set for machine learning. Trained models perform very well in dripping or jetting regimes given (1) a candidate design, (2) any combination of operating requirements, and (3) on generalization tests using small data sets of different fluids by leveraging transfer learning. Since their model performed well in a variety of conditions, they are confident others in the microfluidic community can leverage their web-tool for similar systems. The paper is well written and the analysis is sound. Since the authors make their DAFD tool freely available, this work can make waves in the microfluidic community.

We would like to thank the reviewer for a great summary of the manuscript and detailed suggestions on how to improve the manuscript. We share the same enthusiasm about this tool, and we have revised the manuscript considering your insights. The authors believe the paper has been improved considerably. Please find each of the comments addressed below:

- Given the strong performance of the model and clear execution pathway of their approach, this systematic method to fabricate and evaluate designs can hopefully be implemented in other disciplines beyond the microfluidic community. Since Nature Communications' audience is so broad, the authors should clearly indicate in the manuscript *how* their web tool (or approach to generate the data, and complementary machine learning models) can be adopted in other areas? Can it extend beyond just droplet generation applications in microfluidics? Or can it benefit non-microfluidic applications where rapid prototyping is possible, e.g. designs of 3D printed lattices?

All methods used in the data-set generation and training the machine learning models are application-agnostic. Therefore, not only this can be extended to other microfluidic components, such as micromixers and dielectrophoretic droplet sorters (which the authors are planning to add to the DAFD component library in the future), but also, it can be extended for non-microfluidic applications where accurate performance prediction is lacking and structured data can be generated. However, it should be mentioned that some of the constraints that are added in the design automation step are specific to droplet generation (e.g. if the user specifies a droplet diameter larger than the orifice width, then only the jetting regime will be considered for droplet generation, etc.). To this end, the application-specific constraints in the design automation step are clearly commented in the code-base of DAFD so that researchers could easily adjust or eliminate

these constraints. **To address the reviewers concern in the manuscript we added a paragraph to the discussion on how DAFD framework can be used and extended to other fields:**

The workflow we established in this study is application agnostic, and the authors believe that a similar workflow can be implemented in other microfluidic components such as micro-mixers or even non-microfluidic components such 3D printed lattices to achieve design automation for complex phenomena. Therefore, through design of experiments and rapid-prototyping a structured data-set (conforming to a tabular format) can be generated that can be loaded to DAFD Neural Optimizer to train optimal predictive neural networks. Using these predictive models and by implementing iterative search algorithms such as the one used in DAFD, researchers can achieve accurate design automation. “Page 10”

- The authors work is possible due to a large test bed of data they generated. Also, the authors show how transfer learning makes it possible for others to upload distinct and relatively smaller datasets to DAFD to yield similar results. However, it's not clear how much training data is enough to produce reliable results. It is straightforward to show this via an "ablation study" wherein the machine learning models are trained on progressively larger random subsamples of the overall data and tested against. This way, you could argue concretely that 998 data points were a sufficient starter set. Equivalently, future DAFD users could estimate if their results, e.g. 36 data points, are enough. Essentially, it would be useful to let the readers understand what data collection requirements there are when using DAFD for their applications.

We thank the reviewer for the great comment and suggestions on how to address them. This comment has three prongs: **1-** What is the approximate minimum number of data-points required for training a model from scratch that would result in a similar accuracy to the models trained on all of the data-points? **2-** What is the approximate minimum number of required data-points for accurate transfer learning? **3-** What are the general requirements for data collection in order to extend DAFD? Please see the following responses:

1- We have used the “ablation study” or “data-reduction study” suggested by the reviewer, where we randomly selected 20% (each run a different random 20% is selected, and repeated 10 times) of the bounded dataset (888 data points) as the test-set. Since in the original bounded dataset there are 474 data-points in the dripping regime and 414 data-points in the jetting regime, the size of the randomly selected sub-samples of the training set were incremented as 50, 100, 150, 200, 250, 300, and 325 data-points (the maximum value of subsample train-set size was set to be smaller than the train-set size for jetting regime, i.e., $0.8 \cdot 414 = 331$ data-points). As shown in the figure below, it can be concluded approximately 250-300 data-points for the dripping regime and 200-250 data-points for the jetting regime would yield a relatively similar accuracy. Therefore, for accurate performance prediction and design automation a total of 450-550 data-points can be sufficient to achieve a relatively similar accuracy. It must be mentioned that we used approximately 710 ($0.8 \cdot 888$) data-points for training our models and used the remaining data-points for testing and verification of the models.

To address the reviewers concern in the manuscript we added the ablation study (data-reduction study) results to the main manuscript section 2.2 (Performance prediction) Fig. 4, with accompanying text in the manuscript body and Methods section.

Finally, the effect of the data-set size on the accuracy of the predictive neural networks was studied through a data reduction study. By training the models on increasingly larger sub-samples (starting from 50 data-points up to 325 data-points for each regime) and testing against a 20% randomly selected sub-sample of the original data-set. It can be concluded that approximately 250-300 data-points for the dripping regime and 200-250 data-points for the jetting regime (a total of ~ 500 data-points) would yield a relatively similar accuracy to the full bounded data-set with an 80% train-set (0.8*888 ~ 710 data-points in total), as shown in Fig.4. “Page 5”

2- When using transfer learning, the initial weights and structure of the neural network will be set to the weights and structure of a previously trained neural network trained on a similar system (as opposed to initializing the training process with random weights and structure). Therefore, for a similar system (flow-focusing droplet generation) significantly fewer number of data-points are required to train new accurate models on a new type of fluid. However, an informed approximation cannot be suggested on how many data-points are required for any given new fluid combination. Still, it can be concluded that the closer the fluid properties of the new fluid combination are to the fluid properties of NF 350 mineral oil and DI water the fewer number of data points are required to train a new model using transfer learning. In section 2.3.2 (Transfer Learning) we generated two new small-scale data-sets one on changing the dispersed phase (from DI water to LB bacterial cell media) and another on changing the continuous phase (from NF 350 mineral oil to light mineral oil; viscosity change from 57.2 to 21.4 mPa.s). We demonstrated that transfer learning significantly improves the accuracy of performance prediction (see figure below). With the same number of data-points the models developed using transfer learning performed better for LB bacterial media data-set in comparison to the light mineral oil data-set, which can be attributed to a larger difference in fluid properties in between DI water and LB bacterial media in comparison to

8 Saint Mary's Street
Boston, Massachusetts 02215
T 617-353-2811 F 617-353-7337

difference between NF 350 mineral oil and light mineral oil. Therefore, the larger the changes in the fluid properties of the new fluid combination the greater number of data-points are required for accurate performance prediction.

In this work we quantified the accuracy of our predictive models in Table 1, and showed it was sufficient for accurate design automation. Therefore, researchers can use transfer learning and compare their model performance to the performance benchmark we provided in Table 1, to assess if sufficient data-points are gathered.

To address the reviewers concern in the manuscript we added a new small-scale data-set and quantified the efficacy of transfer learning and compared it to training neural networks from scratch and suggested a pathway to determine the sufficient number of data-points for transfer learning:

“To demonstrate the applicability of transfer learning to performance prediction of droplet generators, we generated two new small-scale data-sets, in which we either changed the dispersed phase fluid or the continuous phase fluid. First, lysogeny broth (LB) bacterial cell media (instead of DI water) and NF 350 mineral oil were used to generate droplets and create a small-scale data-set of 36 data-points, with data-points in both formation regimes. Second, light mineral oil with a viscosity of 21.4 mPa.s with 2% span80 as surfactant (instead of NF 350 mineral oil with a viscosity of 57.2 mPa.s with 5% span80) and DI water were used to create a small-scale data-set of 18 data-points in the dripping regime. Training neural networks from scratch on these data-sets resulted in non-generalizable models that over-fitted to the train-set, and performed poorly on the test-set, as shown in Fig. 5b& c. Conversely, by fine tuning the pre-trained models using transfer

learning, the performance prediction accuracy on the test-set improved significantly (see Fig. 5b&c). Therefore, by using transfer learning only a fraction of initial data-points is required to achieve a comparable accuracy for a new fluid combination. Additionally, transfer learning with 18 data-points per regime showed a higher accuracy for the LB bacterial media data-set in comparison to the light mineral oil data-set. This can be attributed to the smaller difference in fluid properties between LB bacterial media and DI water, in comparison to light mineral oil and NF 350 mineral oil. Therefore, for a new fluid combination that differs significantly in fluid properties (in comparison to NF 350 mineral oil and DI water) more data-points are required for accurate transfer learning. Researchers can use the information provided in Table 1, and compare the performance of their neural networks trained on a data-set (that conforms with the data-set generated in this study in terms of parameter normalization and placement) on new fluid combinations to the performance of the predictive models developed in this study to determine if informative and sufficient number of data-points are gathered.” (Page 7)

3- Generally, if researchers plan to use transfer learning with a relatively small dataset, the data-set should be labeled similar to DAFD dataset (geometric parameters must be normalized to the orifice width, and the definition of flow rate ratio and capillary number should remain consistent) with a similar dataset structure (i.e., column to parameter conversion). On the other hand, if researchers plan to use DAFD Neural Optimizer with a relatively large dataset, they can freely label and structure their dataset (while putting different parameters in the rows and their associated data in columns). **To address the reviewers concern in the manuscript we added:**

“Researchers can use the information provided in Table 1, and compare the performance of their neural networks trained on a data-set (that conforms with the data-set generated in this study in terms of parameter normalization and placement) on new fluid combinations to the performance of the predictive models developed in this study to determine if informative and sufficient number of data-points are gathered.” (Page 7)

- Do DAFD users have to label their data using the same uDROP image-processing software you developed? If they use a different high-speed camera system than what was used in the manuscript, is it safe to pool their data with yours during transfer learning?

When pooling data with DAFD current dataset, how the data is measured is **not** important. u-DROP was developed as an open-source and free image-processing tool for flow-focusing droplet generation, however, DAFD users do NOT have to use u-DROP to add data to DAFD dataset. Additionally, u-DROP has been tested over a wide variety of camera systems (high-speed and non-high-speed cameras), suggesting that u-DROP is independent of the camera setup.

The requirement for safe transfer learning is similar parameter normalization and placement in the tabular form. **We believe we addressed reviewers concern simultaneously as the previous comment.**

- It's great that the DAFD models outperform scaling laws from literature as shown in Figure 3a and 3b. Can the authors compute the same performance metrics in Table 2 for these scaling laws to further aid the comparison?

We thank the reviewer for this comment. We have calculated the quantitative prediction metrics for scaling laws as well.

Scaling law #1 performance in predicting DAFD dataset:

Parameter	Regime	R ²	RMSE	MAPE	MAE
Droplet diameter	Dripping	0.4467	31.7761 um	24.49%	23.3298 um
Droplet diameter	Jetting	-0.6758	58.1142 um	36.29%	46.6100 um
Generation rate	Dripping	0.3499	78.7053 Hz	55.19%	45.8266 Hz
Generation rate	Jetting	-0.2516	117.9551Hz	61.71%	80.4942 Hz

Scaling law #2 performance in predicting DAFD dataset:

Parameter	Regime	R ²	RMSE	MAPE	MAE
Droplet diameter	Dripping	-1.6204	69.1483 um	62.27%	59.3138 um
Droplet diameter	Jetting	0.3724	35.5637 um	22.07%	28.3510 um
Generation rate	Dripping	N.A.	N.A.	N.A.	N.A.
Generation rate	Jetting	N.A.	N.A.	N.A.	N.A.

To address the reviewer's concern, we added both of these tables (Table S3 & S4) to the supplementary information Section 4 (Scaling laws for predicting droplet diameter and generation rate). (page 13 and 14 of the supplementary information)

Reviewer #2 (Remarks to the Author)

- The authors present a design automation tool for generating microfluidic droplet generators based on hydrodynamic flow focusing. The authors made the software tool available as public domain. I regard it as a good manuscript and I think this could develop into a very useful tool in the future that deserves publication in one of the major fluidic journals.

We thank the reviewer for the summary, comments, and suggestions. We believe the manuscript has improved significantly given the valuable feedback provided by this reviewer and the others. The authors envision this manuscript to have a broad range of audience in addition to the microfluidic community. We believe that the large-scale data-set generated, the design automation tool, and the automated online learning framework would interest **machine learning, computer-aided design, and life-science researchers**. Additionally, given that the workflow of DAFD as the first experimentally verified design automation tool in microfluidics was demonstrated in this manuscript, the authors hope that this work would **inspire more researchers in the microfluidic community** to utilize their generated data to develop extensions to DAFD, for it to support additional **fluid combinations, new microfluidic components**, and eventually **sophisticated multi-component microfluidic devices**. This would enable truly modular and functional microfluidic devices. Naturally, as more features are added to DAFD, more researchers will be incentivized to use and contribute to it, acting a positive feedback loop. Analogous to the field of Electronics, and how Electronic Design Automation (EDA) transformed the field, we believe that **design automation could democratize and revolutionize the field of microfluidics** by removing resource-intensive iterative design processes and reducing the expertise required to achieve a functional device. However, this can only be achieved through a collective effort of researchers across several fields including, microfluidics, machine learning, computer-aided design, and life-sciences. Therefore, the authors believe Nature Communications is a suitable venue to show-case an experimentally verified microfluidic design automation framework, and call for a cross-disciplinary collective effort to develop further and more sophisticated Microfluidic Design Automation (MDA) tools.

- Nevertheless, in my view the content of the manuscript is not generic enough and does not have enough novelty and impact to justify publication in Nature communications. Main reasons for that are:

Based on the reviewers' and the editor's feedback we have added **additional experiments, new data-sets with new fluids**, added the new **tolerance study feature**, and revised the manuscript to **emphasize on the generalizability of DAFD** and **highlight the originality of this work** and the potential impact of microfluidic design automation (MDA) tools on the microfluidic, computer-aided design, and life-science communities. We hope to have addressed the reviewer's concerns below:

i) All experiments to train the network are done with water and mineral oil and do not consider changes in surface tension and viscosity of the dispersed or continuous phase. This limits the predictions for real world scenarios employing a wide range of surfactants for the dispensed phase. The authors presented a tool (Neural Optimizer) that allow users to train the network with their own data. Nevertheless, this does not compensate for this weakness in my view.

The authors established the framework of DAFD and verified its accuracy in performance prediction and design automation using a large-scale dataset generated using DI water and NF 350 mineral oil. We also introduced and used the concept of **Transfer Learning** to demonstrate that performance prediction and **design automation can be extended to additional fluid combinations while requiring a significantly fewer number of data-points** as shown below (Figure 1). Therefore, by generating the dataset with DI water and NF 350 mineral oil, we have done the initial heavy lifting to generate a base predictive model (pre-trained model) for flow-focusing droplet generation in a cost-efficient manner. The reagents might not be very exciting, however, without this base model it would be impractical to achieve accurate performance prediction on more expensive/relevant reagents.

FIGURE 1 CONCEPTUAL DEMONSTRATION OF TRANSFER LEARNING IMPLEMENTATION IN DESIGN AUTOMATION OF DROPLET MICROFLUIDICS.

To verify the efficacy of transfer learning, the authors generated **two** new small-scale data-sets on droplet generation. **First** small-scale data-set uses **lysogeny broth (LB) bacterial cell media** (instead of DI water) and mineral oil with only 36 data-points with equal number data-points in dripping and jetting regimes. Second small-scale data-set uses **light mineral oil** (viscosity of 21.4 mPa.s) with **2% Span80** as surfactant, instead of NF 350 mineral oil (viscosity of 57.2 mPa.s) with 5% Span80 with only 18 data-points in the dripping regime.

We demonstrated that transfer learning can be used to extend DAFD either by us or the community to support new fluid combinations with different viscosities and surface tension while requiring a significantly fewer number of data-points see figure below (Figure 2). It was observed that the more different the properties of the new fluid are in comparison to DI water and mineral oil; the greater number of data-points are required to achieve accurate performance prediction and design automation. For example it was observed that with just 18 data-points per regime, transfer learning

was more accurate for the LB bacterial media data-set in comparison to the light mineral oil data-set (where the oil viscosity is less half of the oil viscosity of the original data-set, plus different amounts of surfactant were used).

FIGURE 2 DAFD ENABLES GENERALIZABLE PERFORMANCE PREDICTION.

In addition to the previously established concept of transfer learning, the authors introduced an open-source, web-based tool, **Neural Optimizer**, that allows users to automatically train neural networks on their own large-scale datasets, without requiring an extensive machine learning knowledge. The neural networks generated by Neural Optimizer can be downloaded and easily plugged into the local version of DAFD, to extend its design automation capabilities.

The authors envision that **Transfer Learning** will be used for extending DAFD to support flow-focusing droplet generation with **additional fluid combinations**. Additionally, we envision that **Neural Optimizer** to be used to extend DAFD to support **additional microfluidic components**, such as other methods of droplet generation, droplet sorting, micromixing, cell-trapping, etc. **To address the reviewers concern in the manuscript we added new data-sets that would either change the dispersed phase fluid or the continuous phase fluid and demonstrated that DAFD is generalizable to new fluid combinations and edited the results section 2.3 “Generalizable performance prediction”:**

“To demonstrate the applicability of transfer learning to performance prediction of droplet generators, we generated two new small-scale data-sets, in which we either changed the dispersed phase fluid or the continuous phase fluid. First, lysogeny broth (LB) bacterial cell media (instead

of DI water) and NF 350 mineral oil were used to generate droplets and create a small-scale data-set of 36 data-points, with data-points in both formation regimes. Second, light mineral oil with a viscosity of 21.4 mPa.s with 2% span80 as surfactant (instead of NF 350 mineral oil with a viscosity of 57.2 mPa.s with 5% span80) and DI water were used to create a small-scale data-set of 18 data-points in the dripping regime. Training neural networks from scratch on these data-sets resulted in non-generalizable models that over-fitted to the train-set, and performed poorly on the test-set, as shown in Fig. 5b& c. Conversely, by fine tuning the pre-trained models using transfer learning, the performance prediction accuracy on the test-set improved significantly (see Fig. 5b&c). Therefore, by using transfer learning only a fraction of initial data-points is required to achieve a comparable accuracy for a new fluid combination. Additionally, transfer learning with 18 data-points per regime showed a higher accuracy for the LB bacterial media data-set in comparison to the light mineral oil data-set. This can be attributed to the smaller difference in fluid properties between LB bacterial media and DI water, in comparison to light mineral oil and NF 350 mineral oil. Therefore, for a new fluid combination that differs significantly in fluid properties (in comparison to NF 350 mineral oil and DI water) more data-points are required for accurate transfer learning. Researchers can use the information provided in Table 1, and compare the performance of their neural networks trained on a data-set (that conforms with the data-set generated in this study in terms of parameter normalization and placement) on new fluid combinations to the performance of the predictive models developed in this study to determine if informative and sufficient number of data-points are gathered.” (Page 7)

ii) The approach only covers microfluidic droplet generators based on hydrodynamic flow focusing. Other methods such as droplet generation by step emulsification, droplet generation by T-channel configurations, etc. are not discussed and are not covered in the manuscript.

Although several microfluidic geometries, including T-junction [1], co-flow [2], step-emulsification [3] and flow-focusing [4] are used to produce droplets. Flow-focusing geometries offer a wider range of deliverable performance (diameter and generation rate) [5], [6]. To this end and because flow-focusing droplet generators are more complex from a fluid dynamics standpoint [7], [8], the authors picked flow-focusing droplet generators as a starting point. Given that the authors provided a generic framework for data-set generation, automated training of neural networks, and design automation, we are confident that the other methods of droplet generation can also be learned and design automated, either by us or by the community. However, providing a tool for design automation of all methods of droplet generation is out of the scope of this manuscript and the message it hopes to get across. **To address the reviewers concern in the manuscript we edited the introduction of the manuscript** and added a reasoning of why we chose flow-focusing geometry in this study:

Several geometries including T-junction, step-emulsification, co-flow, and flow-focusing can be used to generate droplets. Flow-focusing geometries offer a wider range of deliverable performance in comparison to the other geometries. Nonetheless, due to the large number of effective parameters and the complex fluid dynamics involved, analytical solutions or generalizable scaling laws are yet to be determined for flow-focusing droplet generation.

(Page 1)

1 Mandatory remarks

1.1- Title

- The claim of the title is too broad and not justified. It should be changed into “Machine Learning Guided Design of Microfluidic Droplet Generators Based on Hydrodynamic Flow Focusing” because other methods than “hydrodynamic flow focusing” (as e.g. droplet generation by step emulsification, T-channels, etc.) are not covered at all in the manuscript.

The authors used the broad “Microfluidic Droplet Generators” term since this tool shows machine learning allows for the design automation of flow-focusing droplet generators for the first time. Flow-focusing droplet generation is one of the more complex forms of droplet generation in terms of the fluid dynamics involved [7]. Since it is expected for machine learning to work similarly in similarly complex systems (given the same amount of data points and data diversity), the authors are confident that machine learning can also provide accurate performance prediction and design automation for other as complex or less complex forms of droplet generation. **To address the reviewers concern in the manuscript we have changed the title of the manuscript to:**

“Machine Learning Enables Design Automation of Microfluidic Flow-Focusing Droplet Generation”.

1.2- Droplet diameter AND droplet frequency as two independent entities

- The authors claim, that the model can predict droplet diameter AND droplet frequency and create the impression that those would be two independent entities. As a “droplet diameter” corresponds to a “droplet volume” and “droplet volume” times “droplet frequency” results in the “flow rate” which is controlled from outside in the experiments of the manuscript, those two are NOT independent entities. So if you control flow rate from the outside, and can predict EITHER “droplet volume” OR “droplet frequency”, you can easily calculate the other entity. The authors keep this impression alive during the whole manuscript as they always give two entities “predicted droplet diameter” and “predicted frequency” and compare those with the outcome of other approaches from the literature (e.g. in figure 3). Either they should justify this explicitly or they should change that.

We would like to thank the reviewer for pointing out this not-intended impression. The authors are aware that droplet diameter, generation rate, and flow rate of the dispersed phase are inter-dependent, and given two of these parameters, the third parameter can be easily calculated. In fact, we **used this concept in section 2.4** of the submitted manuscript (as explained in Methods → Design Automation) to define “**inferred droplet diameter**” and added it to the cost function of the design automation process for a higher accuracy in DAFD.

The main reason the authors have trained different neural networks for diameter and generation separately is to ensure that during the design automation process there is **a redundancy**, in order to **avoid areas** in the design space where **one or both machine learning models are not accurate**. In short, the cost function for design automation consists of three terms, the predicted droplet diameter, the predicted generation rate, and the calculated droplet diameter using the predicted generation rate (which we call: “inferred droplet diameter”). By adding the “inferred droplet diameters” we **entangled the two predictive models** in a way that if the predictions of droplet diameter and generation rate are inconsistent the cost function will increase and the search

algorithm will avoid areas where one or both models are inaccurate (we did not achieve accurate design automation without adding the mentioned redundancy and entangling both machine learning models together, as a control measure). **To address the reviewers concern in the manuscript we edited the manuscript:**

Droplet diameter, generation rate, and flow rate of the dispersed phase are interdependent and given two of the parameters the third parameter can be calculated using the conservation of mass principle as given equation below:

$$\frac{1}{6}\pi D^3 \times F = Q_d,$$

where D is droplet diameter, F is generation rate, and Q_d is the flow rate of the dispersed phase. Therefore, predicting either droplet diameter or generation rate is sufficient to calculate the other. Nonetheless, here we developed separate models for predicting droplet diameter and generation rate to add redundancy in the design automation stage. This enabled defining a new parameter called "inferred droplet diameter" (the diameter calculated using the predicted generation rate and conservation of mass principle), allowing for accuracy-checking of one predictive model using the other predictive model in order to avoid design-spaces where one or both models are inaccurate, as further explained in the design automation section. (Page 2)

- In my view, the fact that both parameters are not independent from each other can also easily be seen in the results. The authors state "DAFD ... showed an average error of 5.41 μm (7.01%) and 38.1 Hz (24.2%)." As the error in the droplet volume scales with the power of 3 on the error in droplet diameter, in my view it is clear from the beginning that an error of $\sim 7\%$ in droplet diameter will result in an error of $\sim 21\%$ in droplet volume and frequency. The same is true for the statement "Droplet diameter prediction was more accurate in comparison to predicting generation rate ..." in the captions of figure 3. In my view this statement is obvious and absolutely not surprising. The authors should clarify to the reader, that droplet diameter and frequency are coupled entities as long as the flow rate is fixed.

The authors are grateful to the reviewer for pointing out this relationship in the average error of droplet diameter and generation rate. The authors agree that the implication of this interdependence on the average error must have been discussed in the submitted manuscript. In fact, given conservation of mass equation we will have:

$$\frac{1}{6}\pi D^3 \times F = Q_w$$

where D is droplet diameter, F is generation rate, and Q_w is dispersed phase flow rate. For a given flow rate we will have:

$$F \sim \frac{1}{D^3} \quad (1)$$

Assuming we can predict D with an error of δ , using the Taylor series expansion we can approximate the error for D^3 with the following equation:

$$\frac{1}{(D + \delta)^3} \approx \frac{1}{D^3} - \frac{3\delta}{D^4} + O^2(\delta)$$

Therefore, the error for generation rate can be approximated as:

$$\delta_f \approx \frac{1}{D^3} - \frac{1}{(D + \delta)^3} \approx \frac{3\delta}{D^4} - O^2(\delta)$$

Dividing both sides by Eq (1) we will have:

$$\epsilon_F = \frac{\delta_f}{F} \approx 1 - \frac{D^3}{(D + \delta)^3} \approx \frac{3\delta}{D} - O^2(\delta)$$

Therefore:

$$\epsilon_F \approx \frac{3\delta}{D}$$

where $\frac{\delta}{D}$ is the percentage error in droplet diameter (ϵ_D). Additionally, for small values of δ , $O^2(\delta)$ will be negligible. Therefore, the percentage error in generation rate (ϵ_F), will be:

$$\epsilon_F \approx 3\epsilon_D$$

The authors are excited to see this relation is valid for our separately trained and different neural networks for droplet diameter and generation rate. As given in Table 1 of the submitted manuscript it can be seen that in the dripping regime the errors are 11.2% (diameter) and 33.5% (generation rate) and in the jetting regime the errors are 4.8% (diameter) and 15.8% (generation rate). Since in this manuscript separate and different neural networks were trained for droplet diameter and generation rate in order to achieve redundancy in design automation (as explained in the response to the previous point of the reviewer). **Two major conclusions can be drawn:**

1- If we train separate models, this error relation will not necessarily be true for every single data-point. However, the error relation will be true when the data-points' errors are averaged together and only if both neural networks are representative of the actual droplet generation phenomena. The fact that **two the separately trained neural networks** with different weights and slightly different structures, are in an agreement with the error relation derived from conservation of mass in both dripping and jetting regimes indicates that the developed machine learning models are **accurate and representative of the actual droplet generation phenomena.**

2- During the design automation validation, DAFD **exceeded the performance of each the predictive neural networks** individually, with an average error of 4.2% (diameter) and 11.5% (generation rate). This **indicates the efficacy of adding the redundancy of performance prediction and entangling** both of the separately trained models in increasing the accuracy of design automation (please refer to the previous point's answer for more information on the redundancy and entanglement of the models). **To address the reviewers concern** in the manuscript we added all the equations for conservation of mass, and percentage error approximation, and the following text to the performance prediction and design automation sections:

The fact that the percentage errors of the separately trained models for droplet diameter and generation rate are compatible with the conservation of mass principle, demonstrates that the models are representative of the droplet generation phenomenon. It must be noted that this error ratio does not necessarily hold true for all of the data-points, further emphasizing the importance of having two separately trained models for diameter and generation rate to enable accuracy-checking of the predictive models, later used in the design automation stage. (Page 4)

Finally, it must be noted that the design automation accuracy exceeded the accuracy of each of the predictive models alone, because of the accuracy-checking (redundancy) in the predictive models

that enabled the introduction of inferred droplet diameter (see Methods: Design automation). (Page 9)

1.3- Comparison with references [15] and [34]

- The authors compare their work with reference [15] and [34] in figure 3. Both references are using a geometrical arrangement of the flow focusing structure, which is similar, but not the same as in the manuscript under review. Continuous phase and dispensed phase are fed in parallel in [15] and [34] while in the current manuscript they meet orthogonally. This should be mentioned and the consequences of it should be discussed. I expect this to make a difference especially for larger flow rates (equivalent to larger frequencies). Finally, as a consequence, I think it is not a fair statement to claim that the new model based on machine learning is better than the scaling models in the two references as the authors did.

This is a great point raised by the reviewer, and the authors agree that the reasoning of how this is a fair comparison should have been discussed more clearly in the submitted version of the manuscript and the supplementary information. We have made changes to the manuscript and supplementary information to address reviewer concerns. We apologize for the relatively long answer in advance and provide our logic for this comparison below:

In the scaling laws geometries, although the oil **inlets** are not exactly placed perpendicular to the water inlet channels, the oil flow **channel** itself becomes perpendicular to the water flow channel upstream of droplet generation (right before the orifice) in both scaling #1 and #2 studies. Therefore, as long as the flow inside the perpendicular section of the oil inlet channel is “**fully-developed**” it can be assumed that the shape of the inlet upstream of the perpendicular section does not affect droplet generation. It is the assumption of a fully-developed flow that allows for any scaling law to be proposed and used, otherwise, the shape of inlets, tubing, syringes, and connectors would invalidate scaling laws. A flow in a channel will be fully-developed after it passes an “**entrance-length**”. The entrance length in a **laminar flow** inside a rectangular channel can be estimated by Eq. (1):

$$L_{ent.} \approx 0.05 Re \cdot D_h \quad (1),$$

where Re is the Reynolds number and D_h is the hydraulic diameter defined by Eq. (2) in rectangular cross-sections:

$$D_h = \frac{4A}{P} = \frac{2WH}{W+H} \quad (2),$$

where W is the oil inlet width and H is the channel depth. Since the Reynolds number is typically less than one in microfluidics, the flow is deemed to be laminar (for our devices the average Re number was less than 0.5). By assuming a large channel dimension of $700 \mu\text{m} \times 700 \mu\text{m}$ (worst-case scenario, i.e., largest oil inlet channel in our study); the hydraulic diameter of the oil inlet channel will be $700 \mu\text{m}$. Additionally, by even assuming a Reynolds number of $Re = 1$, using Eq. (1), the entrance length can be approximated to be less than $35 \mu\text{m}$. Therefore, if the length of the perpendicular section of oil inlet right before the orifice is larger than a couple of tens of micrometers **it is safe to assume the shape of the channels upstream of the perpendicular section of the oil inlet has a minimal effect on droplet generation**. It should be noted that the channel widths and depth used in the scaling law studies, are much smaller than our assumed worst-case scenario, therefore, **the entrance length is expected to be only a couple of micrometers in those studies**.

As shown in Figure 3, the perpendicular section of the oil inlet in the flow-focusing geometry used in scaling law #2 is larger than a couple of tens of microns (not the schematic drawing, but the actual device image). Additionally, since the dimensions used in their study is much smaller than our assumed worst-case scenario, the entrance length is expected to be much smaller than $35 \mu\text{m}$. Therefore, it is safe to assume that the orientation of the channels upstream of the perpendicular section of the oil inlet has a minimal effect on the process of droplet generation.

FIGURE 3 GEOMETRY USED IN SCALING LAW #2 (REF. [34] OF THE SUBMITTED MANUSCRIPT). FIGURES DIRECTLY TAKEN FROM [9].

For scaling law #1, as shown in Figure 4, the perpendicular section oil inlet is longer than the entrance length, therefore, it is safe to assume that the orientation of the channels upstream of the perpendicular section of the oil inlet has a minimal effect on droplet generation.

FIGURE 4 GEOMETRY USED IN SCALING LAW #1 (REF. [15] OF THE SUBMITTED MANUSCRIPT). FIGURES DIRECTLY TAKEN FROM [10] & [11], WHERE [10] IS THE PUBLICATION WHERE THE SCALING LAW WAS TAKEN FROM, AND [11] IS ANOTHER PAPER FROM THE SAME GROUP, SHOWING THE ACTUAL FLOW-FOCUSING GEOMETRY MORE CLEARLY.

Apart from the fluid dynamics, the scaling laws are typically plotted in a log-log scale, which is well in-line with their indented purpose, which is demonstrating the overall effect of major parameters on an output of interest. As shown in Figure 5, when plotting the same data as in Fig. 3 of the main manuscript on a log-log scale, some interesting conclusions can be drawn. First, it can be concluded **as mentioned in the submitted manuscript** the discussed “**scaling laws are great in capturing the overall effect of parameters on performance**”. This can be clearly seen in Figure 5 where both scaling laws successfully captured the overall trend of the DAFD dataset on a log-log scale. Even, when comparing to the data published in the original research articles (scaling law #1 & #2) it can be concluded that the scaling laws show approximately similar accuracies in capturing the overall change in performance, whether using their own dataset or DAFD dataset.

Finally, it can be seen that the scaling laws can actually predicted the performance in DAFD dataset with **higher accuracy at larger flow rates (equivalent to the large generation frequencies)**. Further demonstrating that the fully-developed flow assumption is valid.

FIGURE 5 RECREATION OF FIG. 3 OF THE MAIN MANUSCRIPT IN LOG-LOG SCALE AND COMPARISON OF IT TO FIGURES DIRECTLY TAKEN FROM REF [15] AND [34] OF THE MAIN MANUSCRIPT.

To address reviewer’s concerns all these descriptions and data were added to the supplementary information and a reference to sections was made in the main manuscript:

“Scaling laws for predicting droplet diameter and generation rate” Section 4 of the supplementary information. (Page 11 of Supp. Info.)

1.4- Viscosity and surface tension

- The authors state “In our generated data-set, variations of geometry and flow condition were thoroughly considered. However, fluid properties were kept constant and DI water and mineral oil were used to produce droplets”. Because all experiments are done only with two dedicated fluids, water and oil, I regard the experimental proof of the machine learning approach quite limited. It would be much more interesting if the authors would be able to generate predictions on a wide range of surface tensions (and viscosities) as those parameters change frequently in real world experiments.

The authors agree with the reviewer that the tool will be inherently more useful as the number of fluids it supports increases over-time. However, in the submitted version of the manuscript we demonstrated that DAFD can be extended with transfer learning for accurate performance

prediction of LB bacterial media and NF 350 mineral oil with only 36 data-points (18 data-points per regime).

The authors are delighted that the reviewer also believes a tool that supports microfluidic design automation at multiple fluid level is of major interest. To this end, the authors show-cased two unique solutions to enable extension of DAFD in the future either by the authors or the community.

The 1st approach (Transfer Learning) can be used for extending DAFD to support additional fluid combinations (different surface tension, viscosity, and different surfactants) for droplet generation with flow-focusing geometries. In short, pre-trained neural networks on DI water and mineral oil are used to train new neural networks on new fluids. Since the weights and structure of neural networks will be optimized from a previously established model (as opposed to random initialization), the training process now requires significantly fewer number of data-points. The transfer learning rationale, approach, and validation for using **lysogeny broth (LB) bacterial cell media** instead of DI water (new small-scale data-set 1) and a light mineral oil instead of NF 350 mineral oil (small-scale data-set 2) can be found in section 2.3.2 transfer learning.

The 2nd approach (Neural Optimizer) can be used when there's a need to train models from scratch in an automated fashion. For example, when training neural networks for a new microfluidic component (e.g. electrophoretic droplet sorter, micromixer, etc.) that a pre-trained neural network does not exist, researchers can use Neural Optimizer to train neural networks using this automated and online tool. In section 2.3.1 DAFD Neural Optimizer, we explained the rationale, approach, and validation of the tool.

To address the reviewers concerns in the revised manuscript we added new experiments that would use a **new oil** instead of NF 350 mineral oil and showed the finding of DAFD are generalizable to new fluid combinations. This new oil (viscosity of 21.4 mPa.s) has a viscosity about half of the oil used in the original data-set (viscosity of 57.2 mPa.s). Additionally, we changed the surfactant concentration from 5% in the original data-set to 2% for the new data-set, which changes the surface tension. The results of this new experiments are given in following figure (also discussed in a previous comment)

“To demonstrate the applicability of transfer learning to performance prediction of droplet generators, we generated two new small-scale data-sets, in which we either changed the dispersed phase fluid or the continuous phase fluid. First, lysogeny broth (LB) bacterial cell media (instead of DI water) and NF 350 mineral oil were used to generate droplets and create a small-scale data-set of 36 data-points, with data-points in both formation regimes. Second, light mineral oil with a viscosity of 21.4 mPa.s with 2% span80 as surfactant (instead of NF 350 mineral oil with a viscosity of 57.2 mPa.s with 5% span80) and DI water were used to create a small-scale data-set of 18 data-points in the dripping regime. Training neural networks from scratch on these data-sets resulted in non-generalizable models that over-fitted to the train-set, and performed poorly on the test-set, as shown in Fig. 5b& c. Conversely, by fine tuning the pre-trained models using transfer learning, the performance prediction accuracy on the test-set improved significantly (see Fig. 5b&c). Therefore, by using transfer learning only a fraction of initial data-points is required to

8 Saint Mary's Street
 Boston, Massachusetts 02215
 T 617-353-2811 F 617-353-7337

achieve a comparable accuracy for a new fluid combination. Additionally, transfer learning with 18 data-points per regime showed a higher accuracy for the LB bacterial media data-set in comparison to the light mineral oil data-set. This can be attributed to the smaller difference in fluid properties between LB bacterial media and DI water, in comparison to light mineral oil and NF 350 mineral oil. Therefore, for a new fluid combination that differs significantly in fluid properties (in comparison to NF 350 mineral oil and DI water) more data-points are required for accurate transfer learning. Researchers can use the information provided in Table 1, and compare the performance of their neural networks trained on a data-set (that conforms with the data-set generated in this study in terms of parameter normalization and placement) on new fluid combinations to the performance of the predictive models developed in this study to determine if informative and sufficient number of data-points are gathered.” (Page 7)

Finally, with suggestions from the editor, we added a tolerance study feature to DAFFD. This takes a user-specified tolerance and predicts the deviations in droplet generation performance caused by the tolerances in fabrication or testing. This feature identifies the most sensitive parameters and quantifies how changing those parameters affects the performance in an automated manner. Since, capillary number includes parameters such as surface tension and viscosity, and for a given geometry capillary number changes linearly with oil flow rate. Some tolerances in viscosity and surface tension can be already captured with the developed model on DI water and NF 350 mineral oil. An example of tolerance study report is given below:

8 Saint Mary's Street
 Boston, Massachusetts 02215
 T 617-353-2811 F 617-353-7337

A. Design automation

Specified performance: 100 μm & 200 Hz

B. Predicted performance change with flow rates

C. Predicted droplet diameter tolerance

D. Predicted generation rate tolerance

FIGURE 6 AN EXAMPLE OF TOLERANCE STUDY REPORT GENERATED AUTOMATICALLY WITH USER-SPECIFIED DESIGN AND TOLERANCE (%).

As shown in Figure 6b, we automatically generate plots that show how changing flow rates would affect the observed performance for any given design. Therefore, users can utilize these plots to make informed decision on how to change the flow rates to account for minor changes in viscosity and surface tension. However, for major changes we still recommend using transfer learning and generating few data-points to achieve accurate performance prediction.

A new sub-section called “Design tolerance prediction” was added to the manuscript and further explained in section 2.4.1 (Page 9) Also, this feature was integrated and added to our website.

2- Minor remarks

2.1- I don't think that the case study in single cell encapsulation (section 2.4.1) adds a lot value to the manuscript. In my view it's just a calculator for cell density to be fed to the dispensed phase and this is not a big deal.

The authors agree with the reviewer that the cell density calculation is not based on complex mathematical equations. However, this case-study highlights **two important features of DAFD**.

First, it demonstrates DAFD is still accurate when introducing **design constraints**. By adding two additional design constraints for aspect ratio and water inlet width (as explained in Methods → Single-cell encapsulation) to the desired performance (50 μm & 150 Hz) DAFD proposed a completely different design with different flow rates in comparison to what DAFD previously proposed in section 2.4 (without constraints) for the same performance.

Second, it demonstrates accurate performance prediction enables **additional design automation features** to be added to the existing platform to develop more thorough and sophisticated design automation tools. Eventually, more complex design automation features will be added to DAFD. For example, in droplet merging matching the generation rate of two different droplet generators and maintaining prescribed diameters, while ensuring the average velocity of the two droplet streams are not significantly different to avoid the droplets getting ruptured, once the two streams merge can be challenging and requires design iterations. Another example could be optimizing the design of a device to achieve maximum droplet generation rate, while using the minimum possible amount of oil or optimizing the flow rates of an existing design to achieve smallest possible droplet size or highest possible generation rate. Developing these additional design automation features is possible, however these features are out of scope of this manuscript and the message it hopes to get across. **To address the reviewers concern in the manuscript we rewrote section 2.4.1 (Case-study: Single-cell encapsulation):**

To examine the accuracy of DAFD in providing the required cell concentration to ensure single-cell encapsulation for a user-specified performance, a droplet diameter of 50 μm , a generation rate of 150 Hz, and a λ value (i.e., ratio of cells to droplets) of 0.05 was specified. Additionally, two design constraints were specified in DAFD to test its capability in delivering the desired performance while imposing design constraints. First, the lowest aspect ratio allowable in DAFD (value of 1) was specified as a design constraint to keep the device shallow and maintain the cells in the plane of focus. Second, the lowest normalized water inlet allowable in DAFD (value of 2) was specified as a design constraint to avoid secondary flows inside the inlet channel that could trap the cells.

Naturally, because of the specified design constraints, the geometry suggested by DAFD was different in comparison to the geometry suggested for a diameter of 50 μm , a generation rate of 150 Hz in the design automation section (without design constraints). The suggested device was fabricated and run with the provided flow rates and cell concentration while using 10 μm fluorescent beads as cell surrogates. A droplet diameter of 46.3 μm and a generation rate of 167 Hz was observed (Fig. 8b), demonstrating the efficacy of DAFD in proposing designs with user-specified constraints. Additionally, bead encapsulation rate followed the Poisson distribution closely (Fig. 8c). Slightly higher than expected double-bead encapsulation events were observed, which can be attributed to the weak hydrophobic surface properties of polystyrene that facilitates aggregations of beads suspended in DI water. (Page 9)

2.2- I don't think that reference [8] is adequate to justify the statement „Large machine footprints and high overhead costs limit the accessibility of liquid handling robots [8] ...“

We thank the reviewer for highlighting this. The authors intended point-out the liquid handling robots that can deliver comparable volumes to microfluidics are very expensive and often have large foot-prints. On the other hand, more affordable liquid handling robots often operate on volumes larger than 1 μ L. **To address the reviewers concern in the manuscript we edited the introduction and added the necessary citations**

Large machine footprints and high overhead costs limit the accessibility of high-performance liquid handling robots and cost-effective robots often operate at volumes larger than 1 μ l [8,9].
(Page 1)

2.3- I don't think that reference [14], a paper from 2006, justifies the following statement in year 2020: “... adoption of droplet-based platforms in the life sciences has been an exception rather than the norm [14].” This is especially no true, knowing that digital assays building on microdroplets (Biorad, Stilla, etc.) came up heavily during the last decade. Even the statement there would be a “... lack of predictive understanding [17]” refers to a 10 year old paper and there happened a lot during the last 10 years.

The authors agree with the reviewer that microfluidics in general and droplet-microfluidics specifically have made great progress in the past decade. Additionally, we thank the reviewer for pointing out the relatively old citation. It should be mentioned that commercial droplet-based microfluidic devices such Stilla Naica system for dPCR and 10x Genomics Chromium machine for scRNA-Seq are great success stories that have made it into the market. However, these examples are relatively expensive and designed for a single application thus, the versatility of droplet microfluidics has not reached the market yet. Additionally, as members of life-science institutions such as Boston University's Biological Design Center and Boston University's Medical Campus the authors believe that using microfluidics in life-sciences is still an exception rather than the norm. To this end, the authors envision a reduction in the cost of droplet-based platforms, versatility, and flexibility of its operation (which both can be achieved with proper design automation tools) are necessary in moving towards the full-potential utilization of droplet-microfluidics in life-sciences. We also thank the reviewer, for bringing up the relatively old reference for “the lack of predictive understanding” comment. Unfortunately, the problem has not been solved yet and the community still lacks accurate performance prediction tools for droplet microfluidics. **To address the reviewers concern in the manuscript we edited the introduction with new citations from 2013, 2016, and 2018:**

Adoption of droplet-based platforms in the life sciences has been an exception rather than the norm [15,16]. This can be attributed to the phenomenological complexity of droplet formation [17,18] lack of predictive understanding [19-21], high fabrication cost inherent to photolithography [22], and unreliability of numerical simulations to capture the intricate dynamics of multi-phase flows [21].

(Page 1)

Reviewer #3 (Remarks to the Author):

- This paper describes software to aid the design and simulation of droplet microfluidic devices for forming emulsions. Overall the approach is useful and interesting and should be valuable to the field. The video demonstration is quite useful and should aid adoption of the technology. The paper is well written, concise, and clear, and the plots useful. Overall, the manuscript is thus quite strong.

We thank the reviewer for the positive remarks. The authors believe the manuscript has been significantly improved with **new features** and new **example data-sets**, given the insightful feedback provided by you, the other reviewers, and the editor. We also expect that this manuscript will have a broader impact than just the field of microfluidics, and we hope that our work will be of an interest to the machine-learning, computer-aided design (CAD), design automation, and life-science communities. We refer the reviewer to the responses provided to reviewers #1 & #2, for specifics on the potential broad audience of this microfluidic design automation tool.

- My problem with the manuscript is that I don't think the approach will be that impactful in the field. While indeed generating droplet microfluidic devices is important for research applications, generally speaking the people that build the devices are experts in the design and fabrication. This paper will no doubt aid the design component, but similar devices can already be made with intuition and trial and error. Thus, I do not think the paper will be that impactful to experts.

We thank the reviewer for raising this concern, we added a new feature called “design tolerance” that can provide experts with unique information on the sensitivity of performance to the design parameters for any given design and user-specified tolerance range. Additionally, a performance heatmap for changing the flow rate for any user-specified design can now be generated automatically, to provide a guideline for experts to easily adjust the flow-rates to account for tolerances in fabrication and testing. Additionally, the authors believe that eliminating the need for trial and error can potentially save experts a lot time (and in return cost) and is a major contribution of this work.

Additionally, we humbly would like to point out the potential impact of a microfluidic design automation tool. To this end, we can consider the case of design automation in the field of electronics. Before the introduction of electronic design automation (EDA) printed circuits were designed and laid out manually in a slow, expensive, and error-prone process. It can be argued that designing modern complex integrated circuits (ICs or chips) is practically impossible by hand. Thus, modern IC design would not be imaginable without the specifically developed EDA software. Naturally, without EDA most of the chips taken for granted today would cost too much or simply not exist. EDA started humbly in the 70s, and the introduction of high-level design languages and more features in the 80s allowed for faster, more sophisticated, and bigger designs to be generated more rapidly, better, and simpler.

FIG. 19.24. Drawing a printed circuit. The drawing is carefully planned and executed for photographic reproduction on the circuit board.

*SOURCE: A Manual of Engineering Drawing for Students and Draftsmen, 9th Ed., by French & Vierck, 1960, p. 487.

A few design automation tools exist for microfluidics (mostly place and route tools), however, these tools often model microfluidic networks similar to an IC. Therefore, these tools currently only consider the existence of a flow (existence of a signal or voltage) and do not consider real-world performance dictated by the local flow field. DAFD, is the **first real-world performance-based design automation tool** in the field of microfluidics, and **complements** existing microfluidic design automation tools to achieve design automation based on **high-level performance** specification. DAFD currently supports droplet generation with mineral oil and DI water, however, by developing tools such as Neural Optimizer and using concepts such as transfer learning, we envision DAFD to be extended to **other fluid combinations** either by us or the community (two examples of this extension to droplet generation with LB bacterial cell media & NF 350 mineral and with DI water and light mineral oil were introduced and verified in the manuscript). We also, hope that demonstrated workflow (design of experiments → rapid-prototyping → data → machine-learning → design automation) to be employed by us and the community to extend DAFD to support additional **microfluidic components** (mixers, droplet sorters, mergers, cell-traps, etc.). **To address the reviewers concern** in the manuscript we introduced the **new feature called design tolerance** that would provide new insights for **experts in the field** that was previously impossible without hours of testing and numerical simulations:

A new sub-section called “Design tolerance prediction” was added to the manuscript and further explained in section 2.4.1 (Page 9) Also, this feature was integrated and added to our website. A sample of automatically generated tolerance report is shown below:

8 Saint Mary's Street
 Boston, Massachusetts 02215
 T 617-353-2811 F 617-353-7337

A. Design automation

Specified performance: 100 μm & 200 Hz

1. Orifice width: 150 μm
2. Depth: 300 μm
3. Water inlet width: 375 μm
4. Oil inlet width: 600 μm
5. Outlet width: 300 μm
6. Orifice length: 375 μm
7. Oil flow rate: 6.84 mL/hr
8. Water flow rate: 7.1 $\mu\text{L}/\text{min}$

B. Predicted performance change with flow rates

C. Predicted droplet diameter tolerance

D. Predicted generation rate tolerance

FIGURE 7 DESIGN TOLERANCE REPORT AUTOMATICALLY GENERATED WITH DAFD THAT PROVIDES INSIGHTS ON POSSIBLE PERFORMANCE DEVIATIONS CAUSED BY TOLERANCES IN FABRICATIONS AND TESTING. ALSO, A GUIDELINE IN ADJUSTING THE FLOW RATES TO ACCOUNT FOR THESE TOLERANCES IS PROVIDED.

- For non-experts, this approach only addresses the design challenge. The other challenges for fabricating and successfully operating the devices are still up to the user and, in my opinion, these constitute equal if not even greater challenges. Thus, I do not think the approach will make it substantially easier for non-experts to use such devices. Thus, while the paper is interesting and well done, I am doubtful of the broad impact it will have.

This is a great point raised by the reviewer and authors agree that there are other challenges aside from the iterative, costly, and time-consuming design aspect. The authors also believe that a collective effort made by researchers from several fields on multiple aspects of microfluidics

(fabrication, scalability, surface fouling, mass-manufacturability, turn-key solutions, electronic integration, integrated sensors, cost, etc.) is needed to make microfluidics truly accessible to all, to an extent that it becomes the norm in most life science research laboratory. Naturally, addressing all the problems of microfluidics will be out of the scope of this manuscript or frankly any other manuscript.

However, as mentioned by the reviewer fabrication is another major bottleneck in the accessibility of microfluidic devices. To this end, the current version of DAFD is focused on micromilling polycarbonate sheets as opposed photolithographic/soft-lithographic fabrication of PDMS devices. Although micromilling requires some learning on how use a CNC micromill, this approach is much easier than photolithography, additionally, most of the fabrication is done by the machine itself. Therefore, using a CNC micromill does not require expertise and technical agility (unlike photolithography where most of the work is done extensively manually, and experience and expertise plays a major role in the success rate of fabrication). To this end, DAFD users can use the open in 3DuF button once the design is created, to download design files (.SVG) required for micromilling (as demonstrated in the supplementary video). Finally, we previously characterized and verified a low-cost microfluidic rapid prototyping technique using a low-cost desktop CNC micromill (less than \$2500), that can fabricate microfluidic devices with features as small as 75 μm in less than an hour while costing less than \$10 [12], [13].

Additionally, this work has been partially funded by the DAMP lab (www.damplab.org), a newly formed core facility at Boston University's Biological Design Center. This facility is moving towards becoming a microfluidic service center, where one of the services provided is going to be rapid fabrication of droplet generators for other researcher through DAFD as an interface that would allow researchers to specify their desired performance and possible design constraints. Although this workflow has not been fully implemented yet, DAMP lab is well-poised to provide these services.

To address the reviewers concern in the manuscript we added a new section to the supplementary information called “**An end-to-end workflow**” section S10:

DAFD enables a smooth transition from high-level specification to a working prototype, as shown in Fig. 13. Once a microfluidic flow-focusing design is suggested by DAFD based on the user-specified performance, researchers can use the DAFD-3D μ F module on the website (open in 3D μ F

8 Saint Mary's Street
 Boston, Massachusetts 02215
 T 617-353-2811 F 617-353-7337

button). This would import the suggested design to 3D μ F which an online interactive microfluidic CAD tool. The users can download their design in a format that is compatible with micromilling (.SVG files). These files can be loaded to the software controlling the micromill (e.g. Bantam Tools) to automatically create Gcodes or load the files to CAM software (e.g. Fusion360) for a more advanced control on the Gcodes. Once Gcodes are created the device can be fabricated in a mostly automated manner using a low-cost desktop micromill and assembled as explain is section S1.

References:

- [1] T. Thorsen, R. Roberts, F. Arnold, and S. Quake, "Dynamic pattern formation in a vesicle-generating microfluidic device," *Phys. Rev. Lett.*, 2001.
- [2] C. Cramer, P. Fischer, E. W.-C. E. Science, and undefined 2004, "Drop formation in a co-flowing ambient fluid," *Elsevier*.
- [3] Z. Li, A. Leshansky, L. Pismen, P. T.-L. on a Chip, and undefined 2015, "Step-emulsification in a microfluidic device," *pubs.rsc.org*.
- [4] S. Anna, N. Bontoux, and H. Stone, "Formation of dispersions using 'flow focusing' in microchannels," *Appl. Phys. Lett.*, 2003.
- [5] S. Wiedemeier *et al.*, "Parametric studies on droplet generation reproducibility for applications with biological relevant fluids," *Eng. Life Sci.*, vol. 17, no. 12, pp. 1271–1280, Dec. 2017.
- [6] S. Xu *et al.*, "Generation of monodisperse particles by using microfluidics: Control over size, shape, and composition," *Angew. Chemie - Int. Ed.*, vol. 44, no. 5, pp. 724–728, 2005.
- [7] C. N. Baroud, F. Gallaire, and R. Dangla, "Dynamics of microfluidic droplets," *Lab Chip*, vol. 10, no. 16, p. 2032, 2010.
- [8] P. Korczyk, V. Van Steijn, S. Blonski, ... D. Z.-N., and undefined 2019, "Accounting for corner flow unifies the understanding of droplet formation in microfluidic channels," *nature.com*.
- [9] T. Ward, M. Faivre, M. Abkarian, and H. A. Stone, "Microfluidic flow focusing: Drop size and scaling in pressure versus flow-rate-driven pumping," *Electrophoresis*, vol. 26, no. 19, pp. 3716–3724, 2005.
- [10] W. Lee, L. M. Walker, and S. L. Anna, "Role of geometry and fluid properties in droplet and thread formation processes in planar flow focusing," *Phys. Fluids*, vol. 21, no. 3, p. 032103, Mar. 2009.
- [11] S. Anna, N. Bontoux, and H. Stone, "Formation of dispersions using 'flow focusing' in microchannels," *Appl. Phys. Lett.*, 2003.
- [12] A. Lashkaripour, R. Silva, and D. Densmore, "Desktop micromilled microfluidics," *Microfluid. Nanofluidics*, vol. 22, no. 3, 2018.
- [13] A. Lashkaripour, C. Rodriguez, L. Ortiz, and D. Densmore, "Performance tuning of microfluidic flow-focusing droplet generators," *Lab Chip*, vol. 19, no. 6, 2019.

REVIEWERS' COMMENTS

Reviewer #1 (Remarks to the Author):

The authors have carefully addressed all of my comments satisfactorily with new figures, explanatory text, information in Supplemental, etc.

Reviewer #2 (Remarks to the Author):

I am impressed by the authors, which addressed all the remarks of all three reviewers very carefully and in detail. As a consequence the manuscript improved significantly and is much more mature and solid now. In my view it deserves publication in one of the major microfluidic journals.

Nevertheless, and consistent with reviewer #3, I am still not convinced that it will have enough impact to justify publication in Nature Communications as this journal has a broad readership also beyond microfluidics. One of the leading microfluidic journals in my view would be more adequate.

At the end the tool allows for prediction the droplet diameter for given flow focusing designs at a given flow rate. I doubt that too many experts will change their established design workflows as a result of this manuscript and so I think the impact will be limited.

Of course, I could be wrong about this. So it's the editor's decision to decide if the journal offers the right audience and if the impact of the manuscript will be high enough to justify acceptance in Nature Communications.

Reviewer #3 (Remarks to the Author):

Overall, the authors addressed all the technical elements of the review comments:

- They answer (Reviewer 1) questions with respect to the resolution of data points needed to build accurate models and how many points of "transfer data" the neural model needs to adapt to new regimes, such as fluid types (something also brought up by Reviewer 2). This is laid out as several new figures and paragraphs discussing this adaptive learning module.
- They address the (Reviewer 2) issue of the perceived linkage of droplet diameter and generation frequency by acknowledging their interdependence and discussing how their prediction method is based on comparing predictive results from two different approaches (diameter or frequency centric) as a way to weigh the confidence and accuracy of any output.
- They address the citation concerns of Reviewer 2 by providing additional references and also discussing the nature of citations [15] and [34]. Unfortunately, reported changes for [8,9] are not accurately represented in the manuscript.

Indeed, the major theme of critique from the reviewers with respect to the paper's suitability is that the paper's subject and results are not generalizable enough (in terms of scientific value) to be a Nature Communication paper. Review 1 explicitly states this, Reviewer 2 implies this by saying it would make for a good fluidic journal publication, and Reviewer 3 states that the work here is unlikely to address the fundamental reasons microfluidics is not more widespread.

It feels that the actual advance or breakthroughs of the paper seems to be more at the level of the (neural) machine learning and design framework, and that the microfluidics are just the flavor of "self-designing-the-

perfect-version-of-a-thing" that was focused on. The authors do emphasize that they are more establishing a concept of computer aided design with the response to Reviewer 3, in stating "providing a tool for design automation of all methods of droplet generation is out of the scope of this manuscript and the message it

hopes to get across." Further, the authors claim that the paper is of general interest to machine learning, computer-aided design, and life-science researchers; though, there is very little detail about the actual machine learning theory or mathematics outside of the supplemental information.

One of the biggest author arguments regarding why this is a breakthrough is a comparison of DAFD to the computational tools used to design integrated circuits (ICs) and chips. I personally don't find this a fair comparison, as the device complexity needed of a computer system can be thousands of connection and electronic activity points, with many levels of electronic flow optimization; in contrast, there are few microfluidic applications that naturally, even imaging, would need to be more than 3 or 4 modules before re-interfacing with bulk handling due to the current operational limitations with microfluidics. As such, outside of

a few really custom complicated systems, there is no clear indication that lack of advanced design tools are limiting the field. The authors' further implying their ability to facilitate design maps of micro-milling of microfluidic devices as a solution to fabrication is essentially moot, since micro-milling is an independent solution to the fabrication challenge that has nothing to do with their software or addressing the other half of operation (hands-on) issues that limit microfluidic adoption.

In fact, one could argue, the best solution for enabling microfluidic adoption amongst most life science researchers would be suite of pre-designed cartridges of different parameters that an automated machine runs for them (akin to liquid handling robot automation). Considering that, the author's tools are more suited to helping industry optimize their own devices, rather than anything to do with directly making microfluidics accessible to most users.

Dear reviewers,

We would like to once again thank you for your time and invaluable feedback that helped us improve our manuscript significantly.

Reviewer #1 (Remarks to the Author):

The authors have carefully addressed all of my comments satisfactorily with new figures, explanatory text, information in Supplemental, etc.

We would like to thank the reviewer for their great suggestions and insightful feedback.

Reviewer #2 (Remarks to the Author):

I am impressed by the authors, which addressed all the remarks of all three reviewers very carefully and in detail. As a consequence the manuscript improved significantly and is much more mature and solid now. In my view it deserves publication in one of the major microfluidic journals.

Nevertheless, and consistent with reviewer #3, I am still not convinced that it will have enough impact to justify publication in Nature Communications as this journal has a broad readership also beyond microfluidics. One of the leading microfluidic journals in my view would be more adequate. At the end the tool allows for prediction the droplet diameter for given flow focusing designs at a given flow rate. I doubt that too many experts will change their established design workflows as a result of this manuscript and so I think the impact will be limited.

Of course, I could be wrong about this. So it's the editor's decision to decide if the journal offers the right audience and if the impact of the manuscript will be high enough to justify acceptance in Nature Communications.

We are glad that the reviewer found the changes to the manuscript satisfactory. We would like to once again thank the reviewer for a careful examination of our manuscript which raised great points and helped us improve our manuscript.

The authors are hopeful, that not only the performance prediction, but also, the design automation tool, tolerance prediction feature, the automated online training of neural networks capability, and the open-source data and tool would be of interest to both users and researchers in fields of life-sciences, microfluidics, machine learning, and computer aided design. Additionally, the authors are hopeful that this tool would be the first step towards design automation of droplet generators with more diverse fluid combinations and additional geometries. In addition, the authors believe that with the framework established in this study additional microfluidic components such micro-mixers, droplet sorters, droplet mergers, pico-injectors, etc. can be design automated to create sophisticated microfluidic devices based on high-level performance description. This can lead to more complex tools that can consider operating points and the interactions between different

microfluidic components on the same device to create a functional and optimized multi-component microfluidic device based on high-level description without requiring resource-intensive design iterations.

Reviewer #3 (Remarks to the Author):

Overall, the authors addressed all the technical elements of the review comments:

- They answer (Reviewer 1) questions with respect to the resolution of data points needed to build accurate models and how many points of “transfer data” the neural model needs to adapt to new regimes, such as fluid types (something also brought up by Reviewer 2). This is laid out as several new figures and paragraphs discussing this adaptive learning module.
- They address the (Reviewer 2) issue of the perceived linkage of droplet diameter and generation frequency by acknowledging their interdependence and discussing how their prediction method is based on comparing predictive results from two different approaches (diameter or frequency centric) as a way to weigh the confidence and accuracy of any output.
- They address the citation concerns of Reviewer 2 by providing additional references and also discussing the nature of citations [15] and [34]. Unfortunately, reported changes for [8,9] are not accurately represented in the manuscript.

Indeed, the major theme of critique from the reviewers with respect to the paper’s suitability is that the paper’s subject and results are not generalizable enough (in terms of scientific value) to be a Nature Communication paper. Review 1 explicitly states this, Reviewer 2 implies this by saying it would make for a good fluidic journal publication, and Reviewer 3 states that the work here is unlikely to address the fundamental reasons microfluidics is not more widespread. It feels that the actual advance or breakthroughs of the paper seems to be more at the level of the (neural) machine learning and design frame work, and that the microfluidics are just the flavor of “self-designing-the-perfect-version-of-a-thing” that was focused on. The authors do emphasize that they are more establishing a concept of computer aided design with the response to Reviewer 3, in stating “providing a tool for design automation of all methods of droplet generation is out of the scope of this manuscript and the message it hopes to get across.” Further, the authors claim that the paper is of general interest to machine learning, computer-aided design, and life-science researchers; though, there is very little detail about the actual machine learning theory or mathematics outside of the supplemental information.

One of the biggest author arguments regarding why this is a breakthrough is a comparison of DAFD to the computational tools used to design integrated circuits (ICs) and chips. I personally don’t find this a fair comparison, as the device complexity needed of a computer system can be thousands of connection and electronic activity points, with many levels of electronic flow optimization; in contrast, there are few microfluidic applications that naturally, even imaging, would need to be more than 3 or 4 modules before re-interfacing with bulk handling due to the

current operational limitations with microfluidics. As such, outside of a few really custom complicated systems, there is no clear indication that lack of advanced design tools are limiting the field. The authors' further implying their ability to facilitate design maps of micro-milling of microfluidic devices as a solution to fabrication is essentially moot, since micro-milling is an independent solution to the fabrication challenge that has nothing to do with their software or addressing the other half of operation (hands-on) issues that limit microfluidic adoption.

In fact, one could argue, the best solution for enabling microfluidic adoption amongst most life science researchers would be suite of pre-designed carriages of different parameters that an automated machine runs for them (akin to liquid handling robot automation). Considering that, the author's tools are more suited to helping industry optimize their own devices, rather than anything to do with directly making microfluidics accessible to most users.

We are glad that the reviewer found the authors' response to all technical elements satisfactory. The authors believe references [8] and [9] are presented accurately in the manuscript. In the case of reference [8], the authors mention that "*cost-effective robots often operate at volumes larger than 1 μm* "; which we believe is an accurate representation of what is mentioned in the cited reference as "*One challenge of using our Tecan robot was the minimum allowable volume for reproducible fluid transfer, which was 2 μL for our hardware setup.*". In the case of reference [9], the authors cite this work as a great example of the complex protocols that the digital microfluidic devices can carry out. In reference [9] it was demonstrated that the developed digital microfluidic device can automate an induction optimization assay, which is an extensively manual and labor-intensive process otherwise.

The authors believe that scientific findings of this study are generalizable to other fluid combinations as demonstrated in the results section: "Generalizable performance prediction", with two new data-sets for two new fluid types one for changing the dispersed phase fluid and another for changing the continuous phase fluid. Based on our understanding, reviewer #1 was convinced that the findings are generalizable and asked for more information on how new data-sets should be generated to use the generalizability of the developed tool to even non-microfluidic components such as 3D printed lattices. The authors have made all data-sets and the design automation tool itself open-source and accessible to all to ensure that the findings would be of interest to a broad audience. For instance, the three data-sets generated in this study are available to all researchers in the field of machine learning with interest in microfluidics, to develop more accurate and efficient predictive models for droplet microfluidics using machine learning and transfer learning. Additionally, the source code of the tool is open-source so that the researchers in the CAD (computer aided design) community with an interest in microfluidics can integrate into their already developed CAD tools or future CAD tools (as discussed in the "Discussion" section of the manuscript). The integration of 3D μF software (an online microfluidic CAD tool) and DAFD is an example of how the developed tool can complement other already developed software tools. Also, the automated data-to-model scheme of DAFD Neural Optimizer could be of interest to both life science and microfluidic researchers to create accurate predictive models based on their custom data-sets.

The authors believe the developed tool is a step in the right direction, and its open-source nature allows for the community and us to further develop the tool to add more capabilities and support

additional fluid combinations and microfluidic components. This would pave the way for a much broader audience while reducing the barrier to entry to microfluidics significantly. In addition, the authors believe that with the framework established in this study additional microfluidic components such micro-mixers, droplet sorters, droplet mergers, pico-injectors, etc. can be design automated to create sophisticated microfluidic devices based on high-level performance description. This can lead to more complex tools that can consider different operating points and the interactions between different microfluidic components on the same device to create a functional, optimized, and sophisticated multi-component microfluidic device based on high-level description without requiring resource-intensive design iterations. This would free the researchers from the typical numerous design iterations currently necessary to create a working microfluidic device and enable them to spend time and other resources on developing more complex devices with novel features. Therefore, the authors believe that design automation tools and device complexity go hand in hand, and development of one enables further development of the other, similar to the Electronics industry where design automation tools enabled faster development of integrated chips (ICs), which led to cheaper and more complex ICs.

Finally, the authors would like to point out the connection between the developed design automation tool and low-cost desktop micromilling. The output design of the developed design automation tool is micromill-ready (in SVG format), where the users can use 3D μ F software (an online microfluidic CAD tool) integration module to download the designs as SVG files and directly load the design to a CNC micromill for an easy, rapid, low-cost, and mostly automated fabrication, as demonstrated in the supplementary video. The authors believe that low-cost micromilling is accessible to most researchers for microfluidic fabrication as opposed to the standard photolithography. In fact, it was low-cost desktop micromilling that enabled us to generate the large-scale data-set in a time- and cost-efficient manner in the first place.